# Heat Kernel Goes Topological

## Abstract

We propose a scalable framework for topological deep learning that uses the Heat Kernel Signature (HKS) as a node descriptor on combinatorial complexes. Unlike existing topological neural networks, which rely on expensive higher-order message passing, our approach leverages the Laplacian operator to compute multiscale HKS embeddings that are permutation-equivariant and readily integrated into traditional general deep learning methods. This enables efficient learning while preserving rich structural information. Theoretically, our method achieves maximal expressivity within the limits of spectral techniques, distinguishing complexes up to isospectral equivalence. Empirically, it outperforms existing topological baselines in computational efficiency while maintaining competitive accuracy on molecular property prediction benchmarks. On specialised topological tasks, it shows superior ability to separate complex structures and overcome known blind spots of prior methods. Overall, our results establish HKS-based node features as a powerful primitive for topological representation learning, offering a principled trade-off between expressivity and scalability and opening new directions for applying topological methods in molecular and geometric machine learning.

## 1 Introduction

The intersection of topology and deep learning has unlocked new possibilities for encoding complex structural information in data. Traditional graph-based neural networks have proven effective in learning representations for structured data (Kipf and Welling, 2017; Maron et al., 2019a), yet they often struggle to capture higher-order relationships (Besta et al., 2024). In contrast, combinatorial complexes provide a more expressive framework for modeling such intricate structures, extending beyond pairwise relationships to encode higher-order interactions (Hajij et al., 2022). Current topological neural networks often rely on higher-order message passing protocols to capture complex structural relationships (Verma et al., 2024; Eitan et al., 2025). However, higher order message passing is computationally expensive (cf. Section 5) besides being unable to capture topological and metric invariants (Eitan et al., 2025).

Motivated by these limitations, we propose *TopoHKS*, a scalable and expressive deep learning framework to distinguish non-isospectral combinatorial complexes. Our method leverages Heat Kernel Signatures (HKS) Sun et al. (2009) on combinatorial complexes, yielding continuous, multi-scale descriptors that capture the topological neighbourhood of each cell through a diffusion-based perspective. These descriptors encode rich structural information and serve as input to a transformer network,

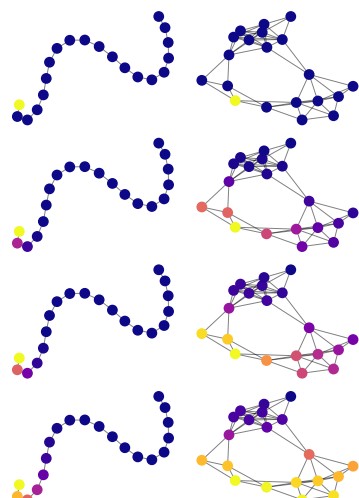

**Figure 1:** Qualitative example of how the Heat Kernel captures local and global structure. We observe how, for two non-isospectral graphs, the 'heat' diffuses differently through the graph.

enabling us to model complex interactions without relying on expensive higher-order message-passing mechanisms. The heat kernel describes how a signal or "heat" would spread over the structure of a combinatorial complex over time (see, e.g., Fig. 1). At very short diffusion times, the heat only reaches a small neighbourhood, capturing fine local structure; over longer times, it spreads further, progressively capturing global connectivity. This natural multiscale behaviour allows us to

| Recent methods for relational data | | | | | Contributions |
|---|---|---|---|---|---|

| Method | SC | R2 | IS | EQ |
|---|---|---|---|---|
| CIN (Hajij et al., 2020) | ✓ | ✗ | ✗ | ✓ |
| SMCN (Eitan et al., 2025) | ✓ | ✓ | ✗ | ✓ |
| MCN (Eitan et al., 2025) | ✗ | ✓ | ✓ | ✓ |
| TopNet (Verma et al., 2024) | ✗ | ✗ | ✗ | ✓ |
| iGN (Cai and Wang, 2022) | ✓ | ✗ | ✗ | ✓ |
| IEGN (Maron et al., 2019b) | ✓ | ✗ | ✗ | ✓ |
| **TopoHKS (ours)** | ✓ | ✓ | ✓ | ✓ |

**Contributions**

**Section 3:**
- Defining the Laplacian for CCs
- HKS descriptor for CC cells

**Section 4:**
- Expressivity of our approach; can distinguish non-isospectral structures

**Section 5:**
- Experiments for expressivity on torus data, scalability, graph and CC classification

**Table 1: Overview of recent methods for relational data and summary of our contributions**. SC: Scalability, R2: Distinguishability above Rank 2, IS: Isospectral, EQ: Equivariance.

encode rich structural information without explicitly simulating high-order message passing, keeping computation efficient. We summarise our contributions below.

1. **Topological embeddings**: We introduce a novel combinatorial complex embedding for deep learning methods that is permutation-invariant and computationally efficient, overcoming limitations in existing approaches.

2. **Theoretical expressiveness**: We establish our framework's fundamental topological and spectral properties, demonstrating its ability to distinguish non-isospectral complexes.

3. **Empirical performance**: Our method achieves state-of-the-art results on molecular prediction benchmarks (s.a. MolHIV, Protein) and topological structure datasets, surpassing existing approaches in distinguishing combinatorial complexes.

4. **Scalability and efficiency**: Unlike traditional higher-order message passing methods, our approach scales efficiently to large combinatorial complexes.

By bridging topological data analysis and deep learning, our work paves the way for expressive, computationally efficient, and theoretically grounded approaches to learning on structured data. As shown in Table 1, we formally introduce our framework, establish its theoretical foundations, and validate its performance through extensive experiments.

## 2 BACKGROUND

**Definition 2.1.** A *combinatorial complex* (CC) is a triple $(S, \mathcal{X}, \mathrm{rk})$ consisting of a set $S$, a subset $\mathcal{X}$ of $\mathcal{P}(S) \setminus \{\emptyset\}$ (where $\mathcal{P}(S)$ denotes the power set of $S$), and a function $\mathrm{rk} : \mathcal{X} \to \mathbb{Z}_{\geq 0}$ with the following properties:

1. For all $s \in S$, $\{s\} \in \mathcal{X}$, and

2. The function $\mathrm{rk}$ is order-preserving, which means that if $x, y \in \mathcal{X}$ satisfy $x \subseteq y$, then $\mathrm{rk}(x) \leq \mathrm{rk}(y)$.

Elements of $S$ are called *entities* or *vertices*, elements of $\mathcal{X}$ are called *relations* or *cells*, and $\mathrm{rk}$ is called the *rank function* of the CC.

This definition was introduced in Hajij et al. (2022). For a combinatorial complex $C$, we write $C^i$ to denote all cells of rank $i$ in $C$. We can also interpret graphs as combinatorial complexes by treating graph nodes as cells of rank 0 and edges as cells of rank 1.

# 3 Heat Kernel Signature Descriptors

Here, we introduce a new approach that defines a Laplacian directly on combinatorial complexes. Next, we construct a heat kernel signature for each node (i.e., 0-dimensional cell), which serves as the input embedding for our deep learning pipeline. We then proceed to describe the whole model setup, subsequently discussing the theoretical and computational benefits of our method in Section 4.

## 3.1 Laplacian on Combinatorial complexes

Laplacians are well-defined operators on graphs (Hein et al., 2007) and geometric shapes (Ovsjanikov et al., 2008). However, extending them to combinatorial complexes introduces several technical challenges not present in graphs or simplicial complexes:

1. **Single-rank connectivity.** Graphs and manifolds each have a single type of connecting element, either edges or surfaces, which directly determines the domain of the Laplacian. In contrast, combinatorial complexes contain connections across multiple ranks, making it less straightforward to specify where and how the Laplacian should operate.

2. **Well-defined hierarchical structure.** In simplicial (chain) complexes, each cell has a fixed rank, and functions are typically defined on cells of a single dimension. The derivative (via boundary or coboundary maps) projects onto adjacent ranks, leading to a natural Hodge Laplacian formulation (Forman, 1998). Combinatorial complexes, however, do not enforce such strict rank stratification, making derivative operations less straightforward to define.

Given these challenges, we aim to design a Laplace operator for combinatorial complexes that is symmetric and positive semi-definite (PSD), converse mass and locality. These are essential properties for Laplacians $L$ on discrete objects, as mentioned in Wardetzky et al. (2007), and ensure that the defined operator is suitable in a heat diffusion setting over time $t$. Since we apply heat diffusion via $e^{-tL}$, we only require $L$ to be PSD. By contrast, Hein et al. (2007) assumes nonnegative weights in their graph construction, which enforces more restrictive constraints on individual entries.

A key aspect of our Laplacian is that it is defined with respect to rank-0 cells, reflecting that downstream tasks typically operate on these base-level elements. This choice ensures that higher-order interactions are captured in their influence on rank-0 cells, with higher-rank cells serving as contextual structures that encode complex dependencies between them.

Given these specifications, we now proceed to construct a Laplacian on combinatorial complexes.

**Definition 3.1** (*Weighted Normalized CC Laplacian*)**.** Let $C$ be a combinatorial complex of maximum rank $R$. For each $i \in \{1, \ldots, R\}$, let $\delta_i \in \{0, 1\}^{n \times m_i}$ denote the incidence matrix between rank-0 cells (vertices) and rank-$i$ cells, $\mathbf{1}$ the vector of size $m$ that contains only ones, and

$$A_i := \delta_i \delta_i^\top - \text{diag}(\delta_i \mathbf{1}) \tag{1}$$

be the adjacency induced by co-membership in rank-$i$ cells (with diagonal entries zeroed out). We also define a set $\mathcal{B}$ of size $R$ such that different subsets have distinct sums; or equivalently, $\forall \mathcal{B}', \mathcal{B}'' \subset \mathcal{B}$ iff $\sum_{b \in \mathcal{B}'} b = \sum_{b \in \mathcal{B}''} b$ than $\mathcal{B}' = \mathcal{B}''$. For instance, we can choose this set to be $\left\{ \frac{2^{-i}}{1-2^{-R}} \mid i = 1, \ldots, R \right\}$, where $R$ is the maximum rank of all CCs in the dataset.

Our aggregated Laplacian is then given by

$$L := \sum_{i=1}^{R} b_i \left( \underbrace{I - D_i^{-1/2} A_i D_i^{-1/2}}_{L_i} \right), \tag{2}$$

where for each rank $i$, its normalized Laplacian $L_i$ - defined using its degree matrix $D_i$, adjacency $A_i$, and identity matrix $I$ - is scaled by the corresponding weight $b_i \in \mathcal{B}$.

Notably, the weight set $\mathcal{B}$ acts as a unique fingerprint of how different ranks contribute to the Laplacian. Distinct subset sums ensure that these contributions remain separable, producing an operator that uniquely characterises each complex (up to isospectral equivalence) while generalising the graph Laplacian to account for higher-order connectivity.

**Definition 3.2.** Two combinatorial complexes are said to be isospectral if their corresponding Laplacian operators have identical spectra (including multiplicities).

## 3.2 Heat Kernels Descriptors on Combinatorial Complexes

Equipped with the CC Laplacian definition, we now construct node descriptors based on HKS.

**Heat kernel on topological structures**    As established in Sun et al. (2009), the heat kernel on a compact manifold $M$ admits the following eigendecomposition:

$$k_t(x, y) = \sum_{i=0}^{\infty} e^{-\lambda_i t} \phi_i(x)\phi_i(y),$$
(3)

where $\lambda_i$ and $\phi_i$ are the $i$th eigenvalue and the $i$th eigenfunction of the Laplace–Beltrami operator, respectively.

For our purpose, instead of the Laplace-Beltrami operator, we appeal to the Laplacian on CCs that we introduced to obtain the following kernel matrix (using matrix exponentiation):

$$K_t = \exp(-tL) \,.$$
(4)

Specifically, the diffusion time $t$ controls the scale of the heat kernel: small $t$ emphasises local neighbourhoods, while large $t$ reveals global connectivity patterns. For efficient computation, we utilise the spectral decomposition of the Laplacian, $L = \Phi^\top \mathrm{diag}(\lambda_1, \ldots, \lambda_n)\Phi$, which yields the following equivalent heat kernel. $K_t = \Phi^\top \mathrm{diag}(e^{-t\lambda_1}, \ldots, e^{-t\lambda_n})\Phi$. Here, $\Phi$ is obtained by arranging the eigenfunctions $\phi_i$ in a matrix.

For a rank-0 cell $c$, we define $K_t(c) := e_c^\top K_t e_c$, where $e_c \in \mathbb{R}^n$ is the basis vector for $c$. Concretely, we obtain these basis vectors by enumerating all rank-0 cells and assigning to each a one-hot encoding of its index. As illustrated in Fig. 1, varying $t$ thus produces a multiscale descriptor that evolves from a local to a global structure. As the heat kernel calculation involves a matrix exponential, we can utilise numerical methods, such as the Nyström Approximation (Li et al., 2010), to accelerate dataset preprocessing.

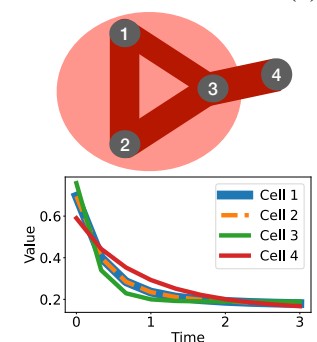

**Figure 2:** Qualitative example of how the HKS descriptor differs for non-isospectral cells of rank 0.

**Definition 3.3** (*Heat Kernel descriptor*). The *descriptor* of a cell $c$ of rank 0 in a combinatorial complex $C$ is a vector in $\mathbb{R}^d$ pertaining to $d$ times, namely, $t_1, \ldots, t_d$. Let $K_t$ be the heat kernel matrix with the variable time parameter $t$. The descriptor is defined as

$$\mathrm{HKS}_{t_1,\ldots,t_d}(c) = [K_{t_1}(c), \ldots, K_{t_d}(c)] \,.$$
(5)

Fig. 2 provides an example, illustrating how the descriptor captures the different topological neighbourhoods of the four cells of rank 0 over time.

## 3.3 Training

The training pipeline is designed to learn a single feature vector that represents the entire combinatorial complex. We begin by computing the Heat Kernel Signature (HKS) for all rank-0 cells, according to Definition 3.3, and concatenate it with any existing features to form enriched input representations. In the appendix, we describe our two approaches: the Transformer and the MLP Mixer. We illustrate the entire pipeline for a classification task in Fig. 3.

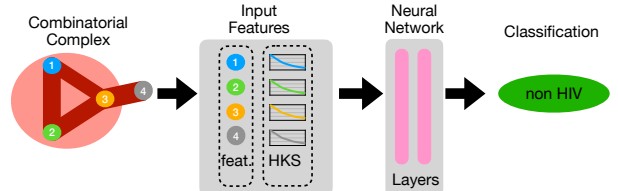

**Figure 3:** The training pipeline. Input: concatenation of cell features of rank 0 and the calculated HKS descriptors. Model: Transformer or MLP Mixer.

# 4 THEORETICAL RESULTS

To support our architectural design and choice of HKS descriptors, we now turn to the theoretical foundations of our framework. We first show that the proposed Laplacian operator satisfies the properties expected of a valid Laplacian and draw connections to the classical Hodge Laplacian. We then analyse the expressive power of our method, demonstrating its ability to capture complex structural information inherent in combinatorial complexes. This section sketches only the key intuition, while the complete proofs are deferred to the Appendix.

## 4.1 LAPLACIAN PROPERTIES

Graphs arise as a special case of combinatorial complexes (CCs) consisting only of rank-0 and rank-1 cells, where each rank-1 cell connects exactly two rank-0 cells. In this setting, the CC Laplacian reduces to the standard graph Laplacian. We first establish that spectrally different combinatorial complexes yield different Laplacians, ensuring that such complexes are distinguishable. We then demonstrate that the natural extension of the Hodge Laplacian—originally defined for cellular complexes—is less expressive on CCs than our proposed operator. Finally, we prove smoothness properties of the CC Laplacian, highlighting its suitability as a foundation for diffusion-based descriptors.

**Corollary 4.1** (*Relationship between CC and Graph Laplacians*). The standard Laplacian for graphs is recovered as a special case of Definition 3.1 when $R = 1$.

The uniqueness requirement for CC Laplacians is established in the next result.

**Theorem 4.1** (*Uniqueness of Laplacians for CCs*). The Laplacian of a combinatorial complex is uniquely determined. Let $L$ be the Laplacian of a combinatorial complex $C$. If there exists an invertible orthogonal matrix $\Pi \in \mathbb{R}^{n \times n}$ such that

$$L' = \Pi L \Pi^\top, \tag{6}$$

then $L'$ serves as the Laplacian of another combinatorial complex $C'$, which is spectrally equivalent to $C$ (i.e., there exists a bijective, unique mapping between the two combinatorial complexes, which makes them isospectral).

The Laplacian is uniquely determined for each combinatorial complex because it is constructed as a weighted sum of adjacency-like components, yielding a spectrum that is unique up to a change of basis. In contrast, as our following result shows, the Hodge Laplacian representation is not unique for CCs whose spectra differ.

**Lemma 4.1** (*Non-uniqueness of Hodge Laplacians on combinatorial complexes*). There exist combinatorial complexes $C_1, C_2 \in C$ such that $C_1 \not\cong C_2$ (i.e., $C_1$ and $C_2$ are not isospectral), yet their corresponding Hodge Laplacians are identical; i.e.,

$$L_H(C_1) = L_H(C_2),$$

where $L_H(C)$ denotes the Hodge Laplacian (Schaub et al., 2020) of the complex $C$.

**Figure 4:** Two combinatorial complexes with their CC and Hodge Laplacian representations. While the CC Laplacians can tell apart the two complexes, the Hodge Laplacian fails to distinguish them.

We illustrate Fig. 4 with two non-isospectral complexes having identical Hodge Laplacians. The distinction arises from a rank-4 cell that connects cell 3 to the rest of the structure via higher-order interactions. While the Hodge Laplacian ignores this cell, the CC Laplacian accounts for all ranks and thus correctly distinguishes the complexes, underscoring its stronger discriminative power.

**Corollary 4.2** (*Hodge Laplacian Expressiveness*). For combinatorial complexes, the Laplacian in Definition 3.1 is strictly more expressive than the Hodge Laplacian. However, both have the same expressivity on simplicial complexes.

While the first part follows directly from the previous lemma, the second statement relies on the structural property of simplicial complexes. Each cell is composed exclusively of cells of one rank lower. As a result, the Hodge Laplacian captures all valid interactions between cells in this setting.

**Laplacian interpretations and implications**

**Theorem 4.2** (*Smoothness*). Let $L$ be the Laplacian descriptor associated with a combinatorial complex $C$, and let $f : C^0 \to \mathbb{R}$ be a function defined on the rank-0 cells of $C$. The quadratic form $f^\top L f$ quantifies the smoothness of $f$ over the complex and can be expressed as

$$f^\top L f = \sum_{i,j} w_{ij} (f_i - f_j)^2 , \tag{7}$$

where $w_{ij}$ is 0 iff there is no cell of any rank connecting the two cells $i$ and $j$.

This connection can be demonstrated by using the definition of the Laplacian, as given in Definition 3.1, and writing out the inner product (details are provided in the Appendix).

It also becomes clear why $f^\top L f$ is a good measure for smoothness. Specifically, the result underscores its role as a discrete Dirichlet energy, quantifying the extent to which $f$ varies across connected cells. Small values of $f^\top L f$ indicate that $f$ changes gradually along connections, implying a smooth signal over the combinatorial complex.

**Remark** (Smoothness of $L$): When interpreting $L$ as a function of the rank, namely, $L(r) := \sum_{i=0}^{r} b_i (I - D^{-\frac{1}{2}} \delta_i \delta_i^\top D^{-\frac{1}{2}})$, we note that $L$ does not change much due to cells of higher rank since the corresponding weights $b_i$ become smaller with rank.

## 4.2 EXPRESSIVENESS OF OUR APPROACH

In this section, we establish the expressiveness of our proposed method via Universal Function Approximation (UFA) Hornik (1991) (formal definition provided in the Appendix). We first show that non-isospectral combinatorial complexes yield different HKS descriptors. Finally, invoking UFA with our CC embeddings, we prove the theoretical expressive power of our approach. Full proofs are deferred to the Appendix, so we present the sketches here.

It is well known that specific neural network architectures satisfy UFA (Hornik et al., 1989). More recently, Transformers (Yun et al., 2020) and MLP-Mixer models Hayase and Karakida (2024) have been shown to possess universal approximation capabilities.

The reasoning underlying our expressiveness results follows an intuitive chain: spectrally distinct combinatorial complexes induce spectrally distinct Laplacians; spectrally distinct Laplacians, in turn, yield distinct heat kernel signatures; and a universal function approximator can learn to separate any distinct HKS descriptors. This implies that, given spectrally different structures, our method can, in principle, learn to distinguish them, even when they differ only in subtle high-rank connectivity.

**Theorem 4.3** (*HKS uniqueness*). Let $L$ and $L'$ be two Laplacians such that $L' \neq \mathbf{\Pi} L \mathbf{\Pi}^\top$ for any orthogonal matrix $\mathbf{\Pi}$. Then the corresponding Heat Kernel Signature (HKS) descriptors derived from $L$ and $L'$ are distinct.

The proof follows two steps.

1. **Uniqueness of the Laplacian:** We first show that if two combinatorial complexes $C$ and $C'$ are not isospectral then their Laplacians $L$ and $L'$ are not similar. That is, there exists no invertible matrix $\mathbf{\Pi}$ such that $L' = \mathbf{\Pi} L \mathbf{\Pi}^\top$.

2. **Diffusion Distinguishability:** Given that the Laplacians are not similar, we then show that their corresponding diffusion patterns (e.g., heat kernels or heat kernel signatures) must differ. This implies that the descriptors derived from diffusion processes can effectively distinguish between non-isospectral complexes.

Having established that different CC have different HKS descriptors, we can now show that the proposed method can distinguish any non-isospectral combinatorial complexes.

**Corollary 4.3** (*Expressiveness*). Let $C$ denote the space of combinatorial complexes, and let $\mathcal{X}$ be the space of HKS descriptors. Suppose $c_1, c_2 \in C$ are two non-isospectral complexes with distinct HKS descriptors $\mathbf{X}_1, \mathbf{X}_2 \in \mathcal{X}$. Then there exists a continuous separating function $g : \mathcal{X} \to \mathbb{R}$ such that

$$g(\mathbf{X}_1) \neq g(\mathbf{X}_2).$$

Moreover, by the universal approximation theorem (Hornik, 1991), such a function $g$ can be approximated arbitrarily well by a suitable neural network $f_\theta$.

**Discussion:** This result is conditional on the ability of HKS to distinguish complexes. While HKS is not a complete isomorphism test—since non-isomorphic isospectral complexes exist—it nevertheless encodes rich structural information. In particular, our method provably enables separating arbitrary combinatorial complexes that differ in their spectrum.

Empirically, we observe (details in the Appendix) that isospectral graphs with different labels typically constitute only a fraction (about $1\% - 3\%$) of real datasets, indicating that such cases are rare in practice. Thus, while our approach does not attain the full isomorphic expressivity (cf. MCN in Eitan et al. (2025)), it nonetheless provides a scalable method that can distinguish the vast majority of combinatorial complexes, regardless of their rank.

### 4.3 SCALABILITY

We consider two backbone architectures for our embeddings: Transformers and MLP-Mixers. While MLP-Mixers offer greater scalability (Tolstikhin et al., 2021), we focus on Transformers for the remainder of our analysis to maintain a concise exposition. The computational complexity of standard Transformers is $O(n^2)$ in the number of input tokens $n$, stemming from the self-attention mechanism. This represents an improvement over SMCNs, whose complexity scales as $O(n^3)$ in the worst case (Eitan et al., 2025). Moreover, since any algorithm that exactly solves graph isomorphism requires exponential time in the worst case, our approach provides a practical trade-off: it distinguishes isospectral complexes within a low-degree polynomial runtime.

## 5 EXPERIMENTS

We now present detailed experiments to demonstrate the empirical benefits of our approach in terms of expressivity and computation. We first show that our method outperforms SMCN on combinatorial complexes that differ in cells of rank at least 3. Moreover, on the torus dataset, we show that our approach can distinguish between all combinatorial complexes that SMCN can separate. We further evaluate our method on established real-world benchmarks.

During the evaluation, we demonstrate that (1) our method outperforms SMCN on most tasks and (2) achieves competitive performance with other baselines on real-world datasets. All models were trained on a single NVIDIA V100 GPU with 32 GB of memory, with training times ranging from 1 to 5 hours depending on the dataset. In total, the project consumed approximately 6000 GPU hours.

In our implementation, we set $d = 10$, i.e., the HKS employs 10 diffusion times, equally spaced between 0 and 3. Across all datasets, we use the same network architecture: a latent dimension of 512 and Transformer-based encoder and decoder modules with 24 self-attention blocks. Owing to the Transformer design, the number of parameters remains fixed as the dataset size increases. Detailed dataset statistics are provided in the Appendix.

### 5.1 TORI AND HIGHER ORDER COMBINATORIAL COMPLEXES.

We start by comparing our method to the SMCN and HOMP on the torus dataset. This dataset is constructed according to the procedure mentioned in (Eitan et al., 2025). Table 2 compares our method with the SMCN and CIN. Notably, our method turns out to be empirically at least as expressive as the SMCN method and also more expressive than CIN on this data.

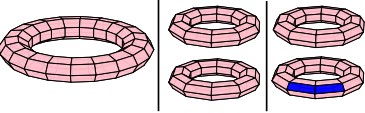

**Figure 5:** Example of a datapoint of the modified torus dataset

We created a new dataset to further compare the expressivity of the two methods. Specifically, this dataset consists of a pair of tori, similar to the original topological blind spot dataset. The only difference is that the tori are different by one cell of rank 4, which covers two cells of rank 2. The proposed method TopoHKS outperformed SMCN in terms of expressivity on this data as well.

We test higher-order expressivity by extending the torus dataset of Eitan et al. (2025). Each sample now includes three complexes: two with the same cover but different structures, and two that differ

| Model | DP | Accuracy | Speed | DP | Accuracy | Speed |
|---|---|---|---|---|---|---|
| SMCN (Eitan et al., 2025) | 0 | 0% | 10 it/s | 223 | 100% | 7 it/s |
| CIN (Bodnar et al., 2021b) | 0 | 0% | 9 it/s | 0 | 0% | 10 it/s |
| TopNetX (Verma et al., 2024) | 0 | 0% | 1 it/s | 0 | 0% | 1 it/s |
| TopoHKS (Hodge) | 0 | 0% | 100 it/s | 223 | 100% | 95 it/s |
| TopoHKS (**this work**) | 223 | 100% | 100 it/s | 223 | 100% | 95 it/s |

**Table 2:** Quantitative and qualitative results for the network to differentiate. The first two columns have the same cover and the second and third columns have the same CC, except one cell of rank 3.

| Model | MolHIV | | PROTEIN | | Glycose | | Immunogenicity | |
|---|---|---|---|---|---|---|---|---|
| | ROC-AUC | speed | ACC | speed | MCC | speed | MCC | speed |
| GCN Kipf and Welling (2017) | 76.06 ± 0.97 | 0.04s | 75.53 ± 1.62 | 0.003s | NaN | NaN | 0.78 ± 0.02 | 20.00 it/s |
| GIN Xu et al. (2018) | 75.58 ± 1.40 | 0.007s | 75.54 ± 1.85 | 0.007s | 0.89 ± 0.02 | 42 it/s | 0.80 ± 0.02 | 44.14 it/s |
| SMCN Eitan et al. (2025) | **81.16 ± 0.90** | 3.1s | 72.8 ± 1.5 | 2.8s | **0.90 ± 0.07** | 3.2 it/s | **0.85 ± 0.01** | 3.3 it/s |
| CIN Bodnar et al. (2021b) | 80.94 ± 0.57 | 3.0s | 77 ± 4.2 | 2.9s | NaN | NaN | NaN | NaN |
| TopNetX Verma et al. (2024) | 75.98 ± 1.80 | 8.4s | 73.79 ± 1.45 | 5.3s | 0.71 ± 0.02 | 0.6 it/s | 0.62 ± 0.04 | 0.5 it/s |
| Transformer | 74.2 ± 0.8 | **0.65**s | 62.2 ± 2.1 | 0.3s | 0.85 ± 0.02 | **4.1**it/s | 0.81 ± 0.03 | **4.0**it/s |
| TopoHKS (Hodge) | 83.0 ± 1.1 | 0.65s | 79.2 ± 0.9 | 0.3s | 0.90 ± 0.01 | 4.1it/s | 0.83 ± 0.00 | 4.0it/s |
| TopoHKS (ours) | **83.1 ± 1.3** | 0.65s | **79.2 ± 1.0** | **0.3**s | **0.90 ± 0.03** | 4.1it/s | 0.84 ± 0.01 | 4.0it/s |

**Table 3:** Performance on Graph Classification (MolHIV and PROTEIN) and simplicial complex datasets (Glycose, Immunogenicity). CIN failed to learn on Glycose and Immunogenicity.

only by the presence of a rank-3 cell. A model succeeds if it separates all three cases. As predicted by theory, existing methods fail on either the same-cover differences or beyond rank-2 cells. Our approach succeeds in both, distinguishing complexes with high rank cells as well as those with the same cover.

## 5.2 COMBINATORIAL COMPLEX LAPLACAIAN EFFECTIVENESS

We observe in Section 5.1 that for CC, where the ranks are discontinous the combinatorial complex laplacian is indeed more powerfull than the Hodge Laplacian. We observe that, when the combinatorial complex is more of an simplicial complex, as it is the case in Table 3 the Hodge Laplacian and CC Laplacian are similar effective. Further we demonstrate that on the evaluated datasets in Table 3 our backbone neural network does by far perform not as well without our additional features. This further supports our thesis that capturing the CC structure with heat diffusion is beneficial.

## 5.3 SCALABILITY COMPARISON

We evaluate the scalability of our method against the Scalable-MCN (SMCN) model from Eitan et al. (2025), using a modified Torus dataset. Instead of classification accuracy, we focus on computational performance as the number of cells increases from 4 to 100. As shown in Fig. 6, SMCN fails to scale beyond 50 cells due to hardware limitations (GPU V100, 32GB). In contrast, our method maintains a constant computational footprint regardless of complex size and benefits from an efficient transformer backbone, achieving higher GPU throughput. On average, it is roughly 12× faster than SMCN.

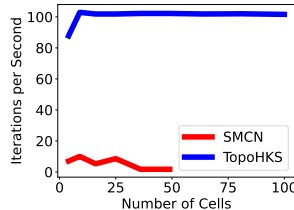

**Figure 6:** Inference timing for differently sized combinatorial complexes. Mean over five runs

## 5.4 GRAPH CLASSIFICATION - BENCHMARKS

Next, we compare our method on established graph classification benchmarks with MolHIV and PROTEIN. The train/val/test setup follows that outlined in Verma et al. (2024). As seen in Table 3, TopoHKS outperforms state-of- the-art topological methods and graph neural networks on these datasets. While the proposed method does not match the lightweight GNNs in terms of computation, its inference time is just a fraction of the established topological methods.

Our findings reveal that with descriptive features, transformers can serve as suitable network architectures for topological learning.

## 5.5 GIFFLAR WITH MLP MIXER

The Gifflar dataset (Joeres and Bojar, 2024) includes naturally higher-order connections and consists of classification tasks: *Glycose* (3 classes) and *Immunogenicity* (2 classes), evaluated using MCC. We train all models using the train/val/test split from Joeres and Bojar (2024), with results shown in Table 3. Our method matches the performance of SMCN while achieving significantly faster inference. It also outperforms both graph neural networks and TopNetX. Notably, CIN fails to learn from these datasets, highlighting the need for expressive yet efficient higher-order neural networks.

## 5.6 MANTRA

We evaluate on Mantra Ballester et al. (2024b) a with the MLP Mixer as the backbone architecture. The results are presented in Fig. 7 When compared against the published results from the benchmark paper, our MLP Mixer achieves an accuracy of 0.93 as a mean, ranking as the 3rd best model overall. This result is particularly noteworthy: many standard graph-based models in the benchmark fail to detect $\beta_1$. Further only 2 other higher order approaches manage to regress well $\beta_1$. The high relative performance of our approach highlights that we are also able to learn regression based tasks as well as that the combinatorial complex based laplacian carries enough information.

## 5.7 EIGENVALUE ABLATION

We validate the efficiency of our spectral approximation of the diffusion pattern on the Immunogenicity dataset. We hypothesize that the task-relevant geometric information is primarily encoded in the low-frequency components of the spectrum. To test this, we conduct an ablation study (results in Fig. 8) by truncating the eigendecomposition to include only the first $k$ eigenpairs, varying $k$ as a percentage of the total number of eigenvalues. Our findings show that the model achieves its full performance using only 10% of the eigenvalues. This demonstrates that the low-frequency spectrum is sufficient to capture the necessary features for this task.

| | | Accuracy | | |
|---|---|---|---|---|
| | | $\beta_0$ | $\beta_1$ | $\beta_2$ |
| DATASET | MODEL (CLASS) | DT | DT | DT |
| 2 - $\mathcal{M}^0$ | MLP | **1.00 ± 0.00** | 0.31 ± 0.00 | 0.92 ± 0.00 |
| | GAT Veličković et al. (2017) | **1.00 ± 0.00** | 0.31 ± 0.00 | 0.92 ± 0.00 |
| | GCN Kipf and Welling (2017) | **1.00 ± 0.00** | 0.31 ± 0.00 | 0.92 ± 0.00 |
| | TAG Du et al. (2017) | **1.00 ± 0.00** | 0.32 ± 0.01 | 0.92 ± 0.00 |
| | UniMP Shi et al. (2020) | **1.00 ± 0.00** | 0.33 ± 0.00 | 0.92 ± 0.00 |
| | CellMP Bodnar et al. (2021a) | 0.46 ± 0.50 | 0.39 ± 0.35 | 0.46 ± 0.44 |
| | CT Ballester et al. (2024a) | **1.00 ± 0.00** | **0.93 ± 0.00** | **0.93 ± 0.00** |
| | DECT Roell and Rieck (2023) | **1.00 ± 0.00** | 0.32 ± 0.00 | 0.92 ± 0.00 |
| | SAN Goh et al. (2022) | 0.09 ± 0.04 | 0.12 ± 0.10 | 0.52 ± 0.14 |
| | SCCN Yang et al. (2022) | **1.00 ± 0.00** | **0.93 ± 0.00** | **0.93 ± 0.00** |
| | SCCNN Yang and Isufi (2023) | 0.00 ± 0.00 | 0.03 ± 0.02 | 0.33 ± 0.37 |
| | SCN Wu et al. (2023) | 0.33 ± 0.38 | 0.21 ± 0.26 | 0.62 ± 0.36 |
| | TopoHKS (ours) | **1.00 ± 0.00** | 0.90 ± 0.00 | 0.89 ± 0.02 |

**Figure 7:** Showcasting the performance on on Mantra dataset

## 6 RELATED WORK

**Node Embedding** Spectral embeddings leverage the graph Laplacian's eigenstructure to capture graph properties (Streicher and Gilboa, 2023; Chung, 1997). Classical methods such as spectral clustering Ng et al. (2001), Laplacian Eigenmaps Belkin and Niyogi (2003), and Diffusion Maps Coifman and Lafon (2006) embed nodes while preserving connectivity. These ideas have influenced modern GNNs that incorporate spectral filters for learning node features (Kipf and Welling, 2017; Fu et al., 2022; Runwal et al., 2022). Unlike persistent homology methods (Hofer et al., 2017), we use Laplacian-based diffusion descriptors with rank-wise expressivity.

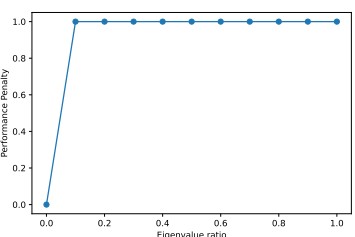

**Figure 8:** Showcasing the Eigenvalue Relationship

**Topological Deep Learning** Topological Deep Learning (TDL) extends learning to higher-order structures, including simplicial, cellular, and combinatorial complexes. Higher-order message passing (HOMP) has been developed for simplicial complexes Bodnar et al. (2021b); Battiloro et al. (2025); Roddenberry et al. (2021) and extended to other complexes Bodnar et al. (2021a); Hajij et al. (2022), boosting MPNN expressivity. Complementary work integrates topological priors into MPNNs Horn et al. (2021); Chen et al. (2021). Benchmarks and architectures are being standardised Telyatnikov et al. (2024); Papillon et al. (2024).

**Expressivity** The expressivity of GNNs is typically measured by their ability to distinguish non-isomorphic graphs, with early results showing equivalence to the 1-WL test (Morris et al., 2019; Xu et al., 2018). More expressive architectures go beyond 1-WL using $k$-WL equivalence Maron et al. (2019b), random features Abboud et al. (2020), subgraph counts Bouritsas et al. (2022), or equivariant polynomials Maron et al. (2019a). Topological models further extend expressivity Zhang et al. (2023); Bar-Shalom et al. (2024), though limitations remain for combinatorial complexes Eitan et al. (2025). For an overview, see Jegelka (2022); Morris et al. (2023).

**Heat Kernel Signatures** Sun et al. (2009) introduced Heat Kernel Signatures (HKS), diffusion-based descriptors capturing multi-scale geometry via Laplace–Beltrami eigenfunctions, with extensions such as WKS Aubry et al. (2011) and refinements for shapes Bronstein and Kokkinos (2010); Ovsjanikov et al. (2010). HKS has also been applied to graphs Donnat et al. (2018), though its role in enhancing GNN expressivity is largely unexplored. Unlike on manifolds with continuous Laplacian operators (Xia and Shi, 2024), we develop discrete rank-aware Laplacians.

**Spectral Kernels and Higher-Order Gaussian Processes** The application of Laplacian-based kernels, such as Radial Basis Functions (RBF) and diffusion kernels, is a well-established paradigm in graph signal processing and regularization Smola and Kondor (2003); Shuman et al. (2013); Bai and Hancock (2004); Borovitskiy et al. (2021). Recent works have successfully extended these spectral techniques to higher-order domains, utilizing functional calculus to define kernels on simplicial and cellular complexes Alain et al. (2023); Yang et al. (2023). Furthermore, multi-scale descriptors derived from spectral graph wavelets and Fourier transforms have been widely adopted to construct powerful kernels for Gaussian Processes (GPs) Opolka et al. (2022; 2023); Zhi et al. (2022); Hammond et al. (2011), with recent extensions specifically targeting topological structures via Hodgelets Alain et al. (2024; 2025). While sharing a spectral foundation with Gaussian Process methods, we instead utilize the Heat Kernel Signature on Combinatorial Complexes as a structural descriptor for a deep learning backbone. This approach bypasses the cubic scaling of GPs and the costs of higher-order message passing, efficiently guaranteeing spectral expressiveness and scalability.

# 7 CONCLUSION

We introduced a framework that integrates the Heat Kernel Signature (HKS) with combinatorial complexes to build expressive, permutation-equivariant representations for deep learning. By defining a Laplacian on combinatorial complexes, we computed multi-scale heat descriptors as robust alternatives to traditional embeddings. Our method is both theoretically expressive, capable of distinguishing non-isospectral complexes, and empirically strong, achieving state-of-the-art results on molecular and topological benchmarks while scaling efficiently to outperform existing methods in runtime. These results underscore the worth of topological descriptors in enhancing graph- and complex-based learning, with a social impact expected to be predominantly positive. In particular, the proposed approach opens up exciting opportunities for topological learning in improving drug design pipelines, brain modelling, etc.

**Future Work** Several directions remain open. Learning heat kernel parameters could improve adaptability across datasets. Extending our method to dynamic combinatorial complexes may enable the study of evolving structures. Finally, combining our approach with contrastive or self-supervised learning could enhance robustness in low-data settings. We expect these steps to strengthen the role of topological deep learning in structured data representation. Furthermore, joint architectures, using our HKS and a message passing approach for isomorphism, will improve the current results.

**Limitations** While the neural network training is efficient and fast per iteration, our method requires an expensive preprocessing step to determine a full eigendecomposition of the Laplacian of each combinatorial complex. This still hinders our method from scaling up to combinatorial complexes to a million cells of rank 0. However, faster approximations of the spectrum can be enabled by well-known techniques such as the Nyström method. Furthermore, our method depends on properly selecting diffusion times and network sizes. Parameterising the features could further improve the performance.

## 8 REPRODUCIBILITY STATEMENT

We include our code in the supplementary material. The repository contains scripts to (i) construct combinatorial complexes and their Laplacians, (ii) compute multi-scale HKS features, and (iii) train/evaluate the Transformer/MLP-Mixer backbones. We provide preprocessing for all datasets used in the paper, along with instructions to reproduce every table and figure (Appendix details the data splits and preprocessing). Our implementation is written in Python/PyTorch, utilising exact hyperparameters as detailed in the Appendix (optimiser, learning rate, batch size, diffusion times, and architecture depth/width). We report results as mean ± std over 5 random seeds (seeds listed in Appendix), and include scripts for deterministic runs where applicable.

We will release the code on GitHub under MIT License upon acceptance. We will also release pretrained checkpoints for the main models to enable direct verification.

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

# 9 APPENDIX

In this part of the Appendix, we fully describe the proofs and provide further definitions if needed. We also include the full text for completeness and ease of reading.

## 9.1 DEFINITIONS

**Definition 9.1** (Universal Function Approximator). A model $f_\theta$ parameterized by $\theta$ is called a *universal function approximator* if, for any continuous function $g : X \to \mathbb{R}^m$ defined on a compact set $X \subset \mathbb{R}^n$ and for any $\varepsilon > 0$, there exists a parameter setting $\theta^*$ such that

$$\sup_{x \in X} \|f_{\theta^*}(x) - g(x)\| < \varepsilon.$$

**Definition 9.2** (Hodge Laplacian). Let $C$ be a cell complex. For $k \geq 0$, denote by $\partial_k : C^k \to C^{k-1}$ the boundary operator, mapping each $k$-cell to its $(k-1)$-dimensional faces. Its transpose $\partial_k^\top$ is the coboundary operator, equivalently the incidence matrix from rank $k$ to $k-1$. The $k$-*th Hodge Laplacian* is defined as

$$\Delta_k = \partial_{k+1} \partial_{k+1}^\top + \partial_k^\top \partial_k. \tag{8}$$

This operator acts on $k$-cochains (real-valued functions on $k$-cells) and captures both upward and downward adjacencies, encoding connections through lower- and higher-dimensional neighbours Hoppe and Schaub (2024).

While this definition extends naturally to combinatorial complexes, their general structure can limit the ability of boundary operators to capture higher-order relationships. As shown in Corollary 9.2, this leads to restricted expressiveness of classical Hodge Laplacians for distinguishing certain complex structures.

**Definition 9.3. Higher order incidence matrix** The incidence matrix represents the relationships between cells of different ranks. Given a combinatorial complex $(S, X, \mathrm{rk})$, the incidence matrix encodes the boundary relationships between cells of consecutive ranks.

Formally, let $X_k$ denote the set of cells in $X$ with rank $k$. The incidence matrix $\delta_k \in \mathbb{R}^{|X_0| \times |X_k|}$ has each entry defined as follows:

$$\delta_k(x, y) = \begin{cases} 1, & \text{if } x \subset y, \\ 0, & \text{otherwise.} \end{cases}$$

This matrix can be interpreted as the **discrete derivative** operator at rank 0 for combinatorial complexes.

**Definition 9.4. Degree Matrix** Let $\delta_k \in \{0, 1\}^{n \times m}$ denote the (unsigned) hypergraph incidence matrix, where $\delta_k(i, e) = 1$ if node $i$ belongs to hyperedge $e$, and zero otherwise. The node degree matrix $D_k \in \mathbb{R}^{n \times n}$ is a diagonal matrix with entries defined as

$$D_k(i, i) = \sum_{e=1}^{m} \delta_k(i, e).$$

A combinatorial complex can be seen as a generalization of a graph. Rank-0 cells correspond to vertices, rank-1 cells to edges, rank-2 cells to filled faces, and so on. Incidence captures "which pieces are part of which higher-rank pieces"—for example, a rank-1 cell (edge) is incident to its two rank-0 cells (vertices), and a rank-2 cell (triangle) is incident to its three rank-1 cells (edges). This hierarchical structure allows us to represent interactions beyond pairs of entities while still relating them back to the base-level vertices.

**Simplicial Complex**

**Definition 9.5** (Simplicial Complex). Let $V$ be a finite set of vertices. A *simplicial complex $K$ over $V$* is a collection of subsets of $V$, called *simplices*, satisfying the following conditions:

1. For every vertex $v \in V$, the singleton $\{v\}$ is in $K$.

2. If $\sigma \in K$ and $\tau \subseteq \sigma$, then $\tau \in K$ (closure under taking faces).

Each element $\sigma = \{v_0, \ldots, v_k\} \subseteq V$ with $|\sigma| = k + 1$ is called a *k-simplex*, and its dimension is defined as $k$.

Examples of such a simplicial complex might include:

- **0-simplices**: vertices
- **1-simplices**: edges
- **2-simplices**: triangles
- **3-simplices**: tetrahedra

A key property of a simplicial complex is that *every subset $\tau \subset \sigma$ of a simplex $\sigma \in K$ must also be* included in $K$. The **dimension** of the complex is the highest dimension among its simplices. Also, each simplex only contains simplices of one lower dimension. For example:

- A graph is a 1-dimensional simplicial complex.
- A triangle mesh is a 2-dimensional complex.

**Definition 9.6** (Cell Complex). A *cell complex $C$* is a topological space constructed inductively from cells:

1. A 0-cell is a point.

2. For $k \geq 1$, a $k$-cell is a space homeomorphic to the open ball $B^k \subset \mathbb{R}^k$.

The complex $C$ is obtained by attaching each $k$-cell $e^k$ to the $(k-1)$-skeleton $C^{k-1}$ via a continuous map

$$\varphi : \partial B^k \to C^{k-1},$$

where $\partial B^k \cong S^{k-1}$ is the boundary of the $k$-ball.

Formally,

$$C^k = C^{k-1} \cup_\varphi e^k,$$

with $C = \bigcup_{k \geq 0} C^k$.

Each $k$-cell is thus attached along its boundary to lower-dimensional cells, generalising simplicial complexes by allowing more flexible cell shapes than simplices.

**Boundary Operator**

**Definition 9.7** (Boundary Operator). Let $K$ be a simplicial complex with vertex set $V$, and let $K_k$ denote the set of $k$-simplices of $K$. The *k-th boundary operator* is the linear map

$$d_k : \mathbb{R}[K_k] \to \mathbb{R}[K_{k-1}], \tag{9}$$

which sends each $k$-simplex to the formal sum of its $(k-1)$-dimensional faces.

Fix an ordering of the vertices in $V$, and represent a $k$-simplex as an ordered tuple $\sigma = [v_0, v_1, \ldots, v_k]$. Then $d_k$ is defined by

$$d_k(\sigma) = \sum_{i=0}^{k} (-1)^i \, \sigma_{-i}, \tag{10}$$

where $\sigma_{-i} = [v_0, \ldots, \widehat{v_i}, \ldots, v_k]$ denotes the $(k-1)$-simplex obtained by deleting the $i$-th vertex from $\sigma$. The signs $(-1)^i$ encode the orientation induced by the chosen vertex order.

The boundary operator reflects how each simplex connects to its lower-dimensional components and is a core concept in *algebraic topology* and *discrete differential geometry*. Those definitions align with the definition from Keros and Subr (2023).

## 9.2 TRAINING SETUP

**Transformer Backbone.**   For the transformer, those features are further enhanced with positional encodings and mapped through a linear embedding layer, producing inputs suitable for different backbone architectures. Let the input be $\mathbf{X} \in \mathbb{R}^{B \times N \times D}$, where $B$ is the batch size, $N$ is the number of rank-0 cells, and $D$ is the feature dimension. Let $\mathbf{G} \in \mathbb{R}^{D \times E}$ be a basis matrix with $E$ the basis dimension. The embedding $\mathbf{O}$ is computed as:

$$\mathbf{O} = \begin{bmatrix} \sin(\mathbf{X} \cdot \mathbf{G}) & \cos(\mathbf{X} \cdot \mathbf{G}) \end{bmatrix}.$$

These encoded features are passed through $n$ layers of self-attention, which enable rich token-level interactions across the combinatorial complex. A final multi-layer perceptron (MLP) aggregates the output into a global feature vector. This design leverages the strong expressivity of attention while preserving permutation equivariance, and an overview is shown in Fig. 3.

**MLP Mixer Backbone.**   As an alternative, we replace the attention layers with an MLP Mixer. In contrast to Transformers, which rely on quadratic-cost self-attention, the MLP Mixer processes spatial and feature information through alternating token-mixing and channel-mixing MLPs. The token-mixing MLPs operate across spatial dimensions (treating tokens as channels), while channel-mixing MLPs act on feature dimensions independently per token. This separation enables efficient modelling of both spatial relationships and feature interactions without the overhead of attention. In our experiments, the MLP Mixer achieves comparable or superior performance to Transformer-based backbones, particularly on tasks requiring global feature interactions, while offering reduced computational cost and faster convergence.

## 9.3 LAPLACIAN PROPERTIES

**Corollary 9.1** (*Relationship between CC and Graph Laplacians*)**.**  The standard Laplacian for graphs is recovered as a special case of Definition 3.1 when $R = 1$.

*Proof.* We show that the combinatorial complex Laplacian $L_C$, when restricted to rank-0 cells and using only rank-1 adjacency, coincides with the standard graph Laplacian $L_G$.

Let $G = (V, E)$ be an undirected graph. Its adjacency matrix $A \in \mathbb{R}^{|V| \times |V|}$ contains 1 when an edge connects nodes. The definition of the graph Laplacian is:

$$L_G = I - D^{-\frac{1}{2}} A D^{-\frac{1}{2}} \tag{11}$$

Now consider a combinatorial complex $C$ consisting only of rank-0 and rank-1 cells, where rank-0 cells correspond to graph vertices and rank-1 cells to edges.

Let $L_C = b_1 \left( I - D_1^{-\frac{1}{2}} A_1 D_1^{-\frac{1}{2}} \right)$ denote the Laplacian on rank-0 cells of $C$. When choosing $b_1 = 1$ then:

$$L_C = I - D^{-\frac{1}{2}} A D^{-\frac{1}{2}} = L_G.$$

Thus, the combinatorial complex Laplacian reduces to the standard graph Laplacian in the rank-0/1 case. □

**Theorem 9.1.** (**Uniqueness of Laplacians for CCs**) Let $L$ be the Laplacian of a combinatorial complex $C$. The Laplacian of a combinatorial complex is uniquely determined. Moreover, if there exists an invertible orthogonal matrix $\mathbf{\Pi} \in \mathbb{R}^{n \times n}$ such that

$$L' = \mathbf{\Pi} L \mathbf{\Pi}^{\top}, \tag{12}$$

then $L'$ serves as the Laplacian of another combinatorial complex $C'$, which is spectrally equivalent to $C$, meaning there exists a bijective, unique mapping between the two combinatorial complexes, which makes them isospectral.

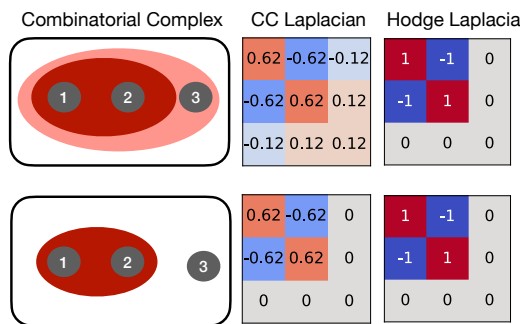

**Figure 9:** Presenting two Combinatorial Complexes with their CC and Hodge Laplacian. While the CC Laplacian differs, the Hodge Laplacian is the same for both complexes.

*Proof.* Let $C$ be a fixed combinatorial complex with a unique set of cells. Since the structure of $C$ is fixed, each incidence matrix $\delta_i$ between rank-0 cells and rank-$i$ cells is uniquely determined (up to orientation and indexing). Consequently, the induced adjacency matrices

$$A_i := \delta_i \delta_i^\top - \text{diag}(\delta_i \mathbf{1})$$

are uniquely fixed by the combinatorial structure.

Consider the construction of the weighted normalized Laplacian

$$L = \sum_{i=1}^{R} b_i \left( I - D_i^{-1/2} A_i D_i^{-1/2} \right),$$

with $b_i \in \mathcal{B}$ drawn from a set of weights satisfying the distinct subset-sum property.

Each $A_i$ is a symmetric positive semidefinite matrix, and any two incidence representations $\delta_i$ and $\delta_i'$ differing only by column permutation or orientation yield the same $A_i$. Thus, $A_i$ is unique up to such trivial transformations. Because the degree matrix $D_i = \text{diag}(A_i \mathbf{1})$ depends only on $A_i$, it is also uniquely determined by the complex.

The use of weights $b_i$ ensures that contributions from different ranks remain separable: if $\sum_{i \in S} b_i = \sum_{j \in T} b_j$ then $S = T$, by construction of $\mathcal{B}$. Hence, the aggregation $\sum_i b_i A_i$ encodes the full rank-wise connectivity pattern without ambiguity.

Finally, normalization by $D_i^{-1/2}$ preserves uniqueness since $D_i$ is itself uniquely determined by $A_i$. Orientation or ordering of higher-rank cells only introduces consistent permutations or orthogonal transformations on $\delta_i$, which cancel out in $A_i$. Therefore, the resulting Laplacian $L$ is uniquely determined up to orthogonal equivalence, and in particular, its spectrum is uniquely defined by the underlying complex. □

**Lemma 9.1.** (**Non-uniqueness of Hodge Laplacians on CC**) Hodge Laplacians on Combinatorial complexes are not unique, meaning there exists a pair of Combinatorial Complexes which share a Hodge Laplacian, but are not isospectral

*Proof.* We prove the corollary by providing a counterexample, as illustrated in Fig. 9. The figure depicts two non-isospectral combinatorial complexes. In both complexes, cells 1, 2, and 3 are of rank 0, and cells 1 and 2 are connected via a rank-1 cell. However, cell 3 is additionally connected to the rest of the structure in the first complex through a higher-order cell of rank 4.

This higher-order connection introduces a structural difference that breaks the isospectralism between the two complexes. The *Combinatorial Complex (CC) Laplacian*, defined in Definition 3.1, captures this distinction by incorporating interactions across all ranks. Specifically, it reflects the influence of the rank-4 cell, which connects otherwise disconnected components at rank 0.

In contrast, the *Hodge Laplacian* fails to distinguish the two complexes. Since the rank-4 cell is not incident to any rank-3 or rank-5 cell, its contribution to the Hodge Laplacian vanishes (as it

produces zero under boundary and coboundary operators). Consequently, the Hodge Laplacians of both complexes are identical.

This example demonstrates that the Hodge Laplacian is not a unique or complete descriptor of combinatorial complex structure. In contrast, the CC Laplacian distinguishes between them, establishing its greater expressiveness and discriminative power. □

**Corollary 9.2.** (**Hodge Laplacian Expressiveness**) On Combinatorial Complexes, the Laplacian in Definition 3.1 is strictly more expressive than the Hodge Laplacian, and on Simplicial Complexes they are equally expressive.

*Proof.* We have shown the first part of the proof in Lemma 9.1.

We now show that the CC Laplacian and the Hodge Laplacian uniquely capture structural differences in cell complexes.

This follows directly from the CC Laplacian's construction: as shown earlier, the Laplacian is uniquely determined by the complex's combinatorial structure. Since the incidence relations between cells are fixed, the CC Laplacian is uniquely defined for any combinatorial complex.

Similarly, the Hodge Laplacian is uniquely defined on cell complexes. First, we observe that the coboundary operator $d_k$, which maps $k$-cochains to $(k + 1)$-cochains, is uniquely determined by the cell structure and chosen orientation. Given this, the Hodge Laplacian,

$$\Delta_k = d_{k-1}^\top d_{k-1} + d_k d_k^\top,$$

is also uniquely defined for each $k$.

Thus, both Laplacians yield unique operators for any fixed cell complex structure. This completes the proof.

□

**Laplacian interpretations**

**Theorem 9.2.** (**Smoothness**) Let $L$ be the Laplacian descriptor associated with a combinatorial complex $C$, and let $f : C^0 \to \mathbb{R}$ be a function defined on the rank-0 cells of $C$. The quadratic form $f^\top L f$ quantifies the smoothness of $f$ over the complex and can be expressed as

$$f^\top L f = \sum_{i,j} w_{ij} (f_i - f_j)^2 , \tag{13}$$

where $w_{ij}$ is 0 iff there is no cell of any rank connecting the two cells $i$ and $j$.

*Proof.* Recall that our Laplacian descriptor is defined as

$$L := \sum_{r=1}^R b_r \left( I - D_r^{-1/2} A_r D_r^{-1/2} \right) = \sum_{r=1}^R b_r L_r, \tag{14}$$

where $A_r$ and $D_r$ denote the adjacency and degree matrices associated with the $r$-th relation, and $b_r \geq 0$ are weights. For any function $f \in \mathbb{R}^n$ defined on the rank-0 cells of $C$, the quadratic form is

$$f^\top L f = \sum_{r=1}^R b_r f^\top L_r f. \tag{15}$$

For each normalized Laplacian $L_r = I - D_r^{-1/2} A_r D_r^{-1/2}$, we have

$$f^\top L_r f = f^\top f - f^\top D_r^{-1/2} A_r D_r^{-1/2} f. \tag{16}$$

Let $g = D_r^{-1/2} f$. Then

$$f^\top L_r f = \sum_u f_u^2 - \sum_{u,v} A_r(u,v) g_u g_v = \frac{1}{2} \sum_{u,v} A_r(u,v) (g_u - g_v)^2, \tag{17}$$

where $g_u = f_u/\sqrt{d_u^{(r)}}$. Substituting back, we obtain

$$f^\top L_r f = \frac{1}{2} \sum_{u,v} A_r(u,v) \left( \frac{f_u}{\sqrt{d_u^{(r)}}} - \frac{f_v}{\sqrt{d_v^{(r)}}} \right)^2. \tag{18}$$

Therefore,

$$f^\top L f = \sum_{r=1}^{R} b_r \cdot \frac{1}{2} \sum_{u,v} A_r(u,v) \left( \frac{f_u}{\sqrt{d_u^{(r)}}} - \frac{f_v}{\sqrt{d_v^{(r)}}} \right)^2. \tag{19}$$

If we absorb the normalization into the weights, define

$$w_{uv} := \sum_{r=1}^{R} b_r \cdot \frac{A_r(u,v)}{2}, \tag{20}$$

then the quadratic form simplifies to

$$f^\top L f = \sum_{u,v} w_{uv} (f_u - f_v)^2, \tag{21}$$

where $w_{uv} \geq 0$ and $w_{uv} = 0$ iff there is no cell of any rank connecting $u$ and $v$. This shows that $f^\top L f$ measures the smoothness of $f$ over the combinatorial complex. □

**Theorem 9.3.** (**HKS uniqueness**) Let $L$ and $L'$ be two Laplacians such that $L' \neq \Pi L \Pi^\top$ for any orthogonal matrix $\Pi$. Then the corresponding Heat Kernel Signature (HKS) descriptors derived from $L$ and $L'$ are distinct.

*Proof.* We assume the uniqueness of the Laplacian $L$, as established in the previous theorem. It remains to show that the corresponding diffusion kernel is uniquely determined by the spectrum of $L$.

Let $L = \Phi \Lambda \Phi^\top$ be the eigendecomposition of the symmetric Laplacian, where $\Phi \in \mathbb{R}^{n \times n}$ is an orthonormal matrix of eigenvectors and $\Lambda = \text{diag}(\lambda_1, \ldots, \lambda_n)$ is the diagonal matrix of eigenvalues. The heat diffusion kernel at time $t > 0$ is defined as:

$$K_t := \Phi \, \text{diag}(e^{-t\lambda_1}, \ldots, e^{-t\lambda_n}) \, \Phi^\top$$

We aim to show that this kernel is unique for a fixed Laplacian. First, note that the exponential function $x \mapsto e^{-tx}$ is strictly decreasing and injective on $\mathbb{R}$. Therefore, the map $\lambda_i \mapsto e^{-t\lambda_i}$ preserves uniqueness of the spectrum.

Since the eigenvectors $\Phi$ are also uniquely determined up to orthogonal transformations (and these cancel in the product $\Phi \Phi^\top$), the matrix $K_t$ is uniquely determined by $L$.

Thus, the diffusion kernel $K_t$ is uniquely defined for a given Laplacian and a fixed diffusion time $t$. If the same kernel were to arise from two distinct spectra $\Lambda \neq \Lambda'$, then we would obtain $e^{-t\lambda_i} = e^{-t\lambda_i'}$ for some $i$, contradicting the injectivity of the exponential map.

Hence, the diffusion pattern is uniquely determined, completing the proof. □

**Corollary 9.3.** (**Expressiveness**) Given two combinatorial complexes with distinct input descriptors, it is possible to learn a function using a Universal Function Approximator (UFA) that effectively distinguishes between them. This means we can determine any combinatorial complexes upto isospectralism by theoretical design.

*Proof.* Let $C_1$ and $C_2$ be two combinatorial complexes. Assume that their node-level input features (e.g., heat kernel signatures) are such that $C_1 \not\cong C_2 \Rightarrow X_1 \not\cong X_2$, i.e., the inputs are distinctive up to isospectralism.

Let $f_\theta$ be a neural network modelled as a Universal Function Approximator (UFA), which takes the input $X$ and computes an output $f_\theta(X)$. Since UFAs can approximate any continuous function to arbitrary precision, there exists a parameterisation $\theta$ such that:

$$f_\theta(X_1) \neq f_\theta(X_2) \quad \text{whenever } X_1 \not\equiv X_2$$

Hence, as long as the inputs have different spectrums, we can obtain distinct representations. This implies that the method is expressive enough to distinguish between combinatorial complexes up to isospectralism. $\square$

## 10  DATASET STATISTICS

**Table 4:** Dataset statistics. Number of graphs, average number of nodes and edges per graph, task type, and percentage of isospectral graphs.

| Dataset | #Graphs | Avg. Nodes | Avg. Edges | Task / Labels | % Isospectral |
|---|---|---|---|---|---|
| Protein (ogbn-proteins) | 1 | 132,534 | 39,561,252 | (112 binary labels) | 3.7 |
| MolHIV (ogbg-molhiv) | 41,127 | 25.5 | 27.5 | Graph binary classification | 1.0 |
| Glycose | 1614 | 115.97 | 452.29 | multi classification | 0.2 |
| Immunogenicity | 1167 | 97.54 | 364.58 | binary classification | 0.4 |

## 11  ABLATION STUDIES

**Table 5:** Ablation study on diffusion times on the Glycose dataset

| Configuration | Accuracy |
|---|---|
| Diffusion times linspace $[0, \ldots, 1]$ | 0.892 |
| Diffusion times linspace $[0, \ldots, 2]$ | 0.897 |
| Diffusion times linspace $[0, \ldots, 3]$ | 0.900 |
| Diffusion times linspace $[0, \ldots, 4]$ | 0.880 |
| Diffusion times linspace $[0, \ldots, 5]$ | 0.919 |

**Table 6:** Ablation study Laplacian weight sets on the Glycose dataset

| Configuration | Accuracy |
|---|---|
| $\mathcal{B} = \{\frac{3}{5}, \frac{2}{5}\}$ | 0.900 |
| $\mathcal{B} = \{\frac{7}{12}, \frac{5}{12}\}$ | 0.899 |
| $\mathcal{B} = \{\frac{3}{8}, \frac{5}{8}\}$ | 0.899 |

Table 5 reports an ablation study on the choice of diffusion times for the Glycose dataset. Extending the diffusion horizon generally improves accuracy, with the highest value of 0.919 achieved when using times evenly spaced between 0 and 5. The results are not strictly monotonic, as the range up to 4 performs worse than shorter ranges, showing that the diffusion scale influences the balance between local and global information.

Table 6 presents results for different Laplacian weight sets. Performance remains stable at approximately 0.899–0.900 across all tested configurations, indicating that the model is robust to the precise choice of weights as long as the distinct subset-sum property holds. These findings suggest that diffusion times are a more decisive factor for performance than weight selection in this setting.

## 12 Eigendecomposition and Acceleration

We did not explore acceleration strategies for our preprocessing pipeline, since computing descriptors for 1000 combinatorial complexes from the Gifflar Glycose dataset requires only about 20 minutes, whereas training the model itself takes approximately 1.5 hours. Thus, the preprocessing time is significantly shorter and not the main bottleneck. A further limitation is the lack of available datasets containing higher-order large-scale graphs, which prevents systematic benchmarking in this regime. In fact, no existing topological method is capable of handling larger combinatorial complexes, meaning that although our approach could, in principle, process them, we cannot compare against other baselines. Developing datasets and methods that enable scaling to larger complexes, therefore, remains an interesting direction for future work.

