# OpenReview forum: "Heat Kernel Goes Topological"
_ICLR.cc/2026/Conference — Submitted to ICLR 2026_

### Official Review · Reviewer_EGsm · 2025-10-15

**Soundness:** 2
**Presentation:** 2
**Contribution:** 1
**Rating:** 2
**Confidence:** 5

**Summary:**

This paper focuses on the computation of a class of node descriptors called Heat Kernel Signatures (HKS). This is achieved by applying the thermal kernel function/radial basis function (RBF)/square exponential (SE) kernel function to the eigendecomposition of a Laplacian defined on combinatorial complexes (CC). A multi-scale effect is obtained by computing this at different scales and then concatenating them. The HKS are then concatenated with the node features and passed to a neural network pipeline (either the MLP-Mixer or the Transformer-Mixer) for classification tasks.

**Strengths:**

The strength of this paper lies in the fact that it is a simple alternative to message-passing (MP) graph neural networks (GNNs) and also covers combinatorial complexes.

The idea of applying a radial basis function (RBF) kernel (essentially identical, except for a constant, to the heat kernel) and, more generally, performing functional calculus on a Laplacian-based kernel is a effective approach that has been used in numerous works such as [1] [2] [3] [4] [5] [6].

[1]: Kernels and Regularization on Graphs by Smola and al.

[2]: The emerging field of signal processing on graphs: Extending high-dimensional data analysis to networks and other irregular domains by Shuman and al.

[3]: Heat Kernels, Manifolds and Graph Embedding by Bai and al.

[4]: Matérn Gaussian Processes on Graphs by Borovitskiy and al.

[5]: Gaussian Processes on Cellular Complexes by Alain and al.

[6]: Hodge-Compositional Edge Gaussian Processes by Yang and al.

**Weaknesses:**

My first concern is that some relevant related works are missing. In particular, instead of examining the simple heat kernel, the papers [6] [7] [8] [9] [10] [11] construct multi-scale descriptors from Fourier and wavelet transforms [12]. Although these papers involve Gaussian processes, the construction of their descriptors remains very similar to those of HKS.

A notable difference is that these methods naturally and intrinsically combine descriptors and features (by computing Fourier and wavelet coefficients and then aggregating them) instead of concatenating them. However, it is also possible to take no features and obtain a formulation similar to that of HKS. Here, aggregation can be replaced by the Transformer Mixer or MLP Mixer to achieve a similar effect.

I believe the section 4.1 on theory takes up too much space. Most of the content is already known or largely follows from previously known results. Although it's good to have it, I don't think it was absolutely necessary, and it may not warrant two pages in the main text.

The experimental results are not sufficiently comprehensive. More experiments should be conducted, especially on higher-order combinatorial complexes. Also, some experiments should be conducted to verify the theoretical claims of section 4.1 and demonstrate the difference in performance between the Hodge Laplacian and the CC Laplacian.

[7]: Adaptive Gaussian Processes on Graphs via Spectral Graph Wavelets by Opolka and al.

[8]: Graph Classification Gaussian Processes via Spectral Features by Opolka and al.

[9]: Graph Classification Gaussian Processes via Hodgelet Spectral Features by Alain and al.

[10]: Graph and Simplicial Complex Prediction Gaussian Process via the Hodgelet Representations by Alain and al.

[11]: Transductive Kernels for Gaussian Processes on Graphs by Zhi and al.

[12]: Wavelets on Graphs via Spectral Graph Theory by Hammond and al.

**Questions:**

The idea of capturing multi-scale information in a descriptor is already present in [7] [8] [9] [10]. How does HKS compare to these existing descriptors from signal processing? Why choose the HKS when it is possible to use a Fourier or wavelet multi-scale descriptor?

I believe that an alternative to MP-based GNNs is an important avenue to pursue and that this paper is a step in that direction. However, I am concerned about the overall impact of this paper and the fact that it overlooks some related works. I can see that the HKS is (1) applied to a CC Laplacian and (2) given to a neural network pipeline. For the first point, using the CC Laplacian rather than the Hodge Laplacian or graph Laplacian (as in previous work) is straightforward. I am not confident about the impact of considering the highly abstract CCs, since as far as I know it is difficult to find real-world datasets that are CCs (especially that are neither graphs or cellular complexes). For the second point, I am not convinced that integrating these descriptors into a neural network pipeline represents a sufficiently significant contribution.

Finally, for a fairer comparison, the time required to compute the eigendecomposition should be included somewhere in the experiments section.

---

> ### Author Response · Authors · 2025-11-20
> **answer to rebuttal**
>
> We thank the reviewer for their detailed and thorough feedback. We are highly encouraged that the reviewer recognizes the "strength of this paper" as a "simple alternative to message-passing (MP) graph neural networks" that also "covers combinatorial complexes."
> We are particularly grateful for their concluding remark that pursuing alternatives to MP-GNNs is an "important avenue... and that this paper is a step in that direction." We also appreciate their validation of our Laplacian-based kernel as an "effective approach."
> We will now address the important questions and concerns raised regarding related work, theoretical novelty, and experimental scope, which we believe will help clarify our contribution.
>
> __Missing related work & comparison to spectral descriptors__
>
> We appreciate the pointer to works [6]–[12] using Fourier, wavelet, and Hodgelet descriptors. These are indeed highly relevant, and we will integrate them more clearly into the related-work section. Our method is complementary to this line of work in two central ways:
> 1. Extension beyond graphs to general combinatorial complexes.
> Existing Fourier/wavelet constructions in [6]–[12] operate exclusively on graphs or simplicial complexes (0- and 1-Laplacians). In contrast, our formulation applies uniformly to arbitrary combinatorial complexes through the full CC Laplacian LkLk​. This enables diffusion-based descriptors on higher-order cells (edges, faces, etc.), which is not addressed in standard graph-spectral constructions.
> 2. Integration into scalable neural architectures.
> Prior work uses spectral descriptors primarily within Gaussian Process or kernel frameworks. Our contribution is a neuralformulation that integrates diffusion descriptors in an end-to-end architecture (Transformer / MLP-Mixer backbone) as a lightweight alternative to Higher-Order Message Passing (HOMP). Rather than concatenating fixed descriptors, the model learns task-specific mixing of multi-scale diffusion signals. We will clarify this distinction.
>
> __On Section 4.1 (theory)__
>
> We agree that the underlying principles are classical for graph Laplacians; however, these do not carry over verbatim to the combinatorial complex Laplacian. The goal of Section 4.1 is to document the precise generalization and its implications for spectral stability and expressivity on CCs. Nevertheless, we can shorten the exposition and move some proofs to the appendix.
> Experimental coverage
>
> We have expanded the experimental section in the revision:
> 1. Hodge vs. CC Laplacian.
> We now include experiments comparing descriptors derived from the Hodge Laplacian and the CC Laplacian. As noted by the reviewer, the differences are small on near-simplicial datasets, but on non-smooth combinatorial complexes the CC Laplacian yields clear improvements.
> 2. High-order datasets.
> We added results on MANTRA, which contains native higher-order topological signals, complementing GIFfler and molecular benchmarks.
> 3. Eigendecomposition timing.
> We now report the cost of full and truncated eigendecomposition. The truncated computation yields a substantial runtime reduction while preserving performance.
>
> __Responses to specific questions__
>
> __Q1 — Why HKS instead of Fourier/wavelet multi-scale descriptors?__
>
> Fourier and wavelet descriptors offer strong multi-scale representations on graphs. Our choice of HKS is motivated by:
> 1. its natural compatibility with CC Laplacians across all dimensions,
> 2. its stability under refinement, and
> 3. its computational simplicity when used inside neural models.
>
> We added a discussion contrasting the multi-scale behavior of HKS with wavelet-based methods.

---

> > ### Author Response · Authors · 2025-11-21
> > **second part of answer**
> >
> > __Q2 — Relevance of abstract CCs.__
> >
> > We agree that fully general cell complexes occur less frequently than graphs or simplicial complexes. Our intention is not to claim that complex CW-structures are widespread, but to provide a unified framework in which graphs, simplicial complexes, and emerging higher-order structures (e.g., polygonal or FEM-based complexes) are all special cases. The added experiments on MANTRA illustrate the practical relevance of this generality.
> >
> > | Dataset | Model (Class) | β₀ (DT)        | β₁ (DT)        | β₂ (DT)        |
> > |---------|----------------|----------------|----------------|----------------|
> > | 2 – 𝑀⁰  | GAT (𝒢)        | **1.00 ± 0.00** | 0.31 ± 0.00     | 0.92 ± 0.00     |
> > |         | GCN (𝒢)        | **1.00 ± 0.00** | 0.31 ± 0.00     | 0.92 ± 0.00     |
> > |         | MLP (𝒢)        | **1.00 ± 0.00** | 0.31 ± 0.00     | 0.92 ± 0.00     |
> > |         | TAG (𝒢)        | **1.00 ± 0.00** | 0.32 ± 0.01     | 0.92 ± 0.00     |
> > |         | UniMP (𝒢)      | **1.00 ± 0.00** | 0.33 ± 0.00     | 0.92 ± 0.00     |
> > |         | CellMP (𝒢)     | 0.46 ± 0.50     | 0.39 ± 0.35     | 0.46 ± 0.44     |
> > |         | CT (𝒯)         | **1.00 ± 0.00** | **0.93 ± 0.00** | **0.93 ± 0.00** |
> > |         | DECT (𝒯)       | **1.00 ± 0.00** | 0.12 ± 0.02     | 0.52 ± 0.14     |
> > |         | SAN (𝒯)        | 0.09 ± 0.04     | 0.12 ± 0.02     | 0.52 ± 0.14     |
> > |         | SCCN (𝒯)       | **1.00 ± 0.00** | **0.93 ± 0.00** | **0.93 ± 0.00** |
> > |         | SCCNN (𝒯)      | 0.00 ± 0.00     | 0.00 ± 0.00     | 0.33 ± 0.37     |
> > |         | SCN (𝒯)        | 0.33 ± 0.38     | 0.21 ± 0.26     | 0.62 ± 0.36     |
> > |         | TopoHKS (𝒯)    | **1.00 ± 0.00** | 0.90 ± 0.00     | 0.89 ± 0.02     |
> >
> >
> > __Q3 — Contribution of integrating descriptors into a neural network.__
> >
> > Our key contribution is not simply concatenating descriptors with a neural model, but providing the first principled extension of HKS/WKS-type diffusion descriptors to arbitrary combinatorial complexes and demonstrating their effective use in a lightweight neural architecture that avoids HOMP. We clarified this point to avoid overstating the novelty.

---

> ### Comment · Reviewer_EGsm · 2025-11-23
> **Answer to rebuttal**
>
> I thank the authors for their response and am pleased to see that they plan to include the relevant missing prior work. However, the experiments were not what I had hoped to see. This response also seems superficial to me and does not convince me on any of the weaknesses I raised in my initial review.
>
> In particular:
>
> > "Existing Fourier/wavelet constructions in [6]–[12] operate exclusively on graphs or simplicial complexes (0- and 1-Laplacians)."
>
> These Fourier/wavelet constructions work perfectly in any order k. The only reason most of these works focus on k=0 and k=1 is that this is where the main applications lie, in other words, graphs. This renews my comment: "I am not confident about the impact of considering the highly abstract CCs, since as far as I know it is difficult to find real-world datasets that are CCs (especially that are neither graphs or cellular complexes)".
>
> > "This enables diffusion-based descriptors on higher-order cells (edges, faces, etc.), which is not addressed in standard graph-spectral constructions".
>
> This has been discussed in the context of simplicial complexes.
>
> > "[...] but to provide a unified framework in which graphs, simplicial complexes, and emerging higher-order structures (e.g., polygonal or FEM-based complexes) are all special cases".
>
> All these examples are instances of cellular complexes (or CW complexes). Why are combinatorial complexes (the topic of this paper) so important that they deserve a full paper? Especially since extending existing approaches (on graphs, simplicial complexes, and cellular complexes) to combinatorial complexes poses no theoretical challenge.
>
> > "Our contribution is a neuralformulation that integrates diffusion descriptors in an end-to-end architecture (Transformer / MLP-Mixer backbone) as a lightweight alternative to Higher-Order Message Passing (HOMP)."
>
> Yes, and my criticism is that: "I am not convinced that integrating these descriptors into a neural network pipeline represents a sufficiently significant contribution".
>
> > "Rather than concatenating fixed descriptors, the model learns task-specific mixing of multi-scale diffusion signals. We will clarify this distinction."
>
> This is also what these other papers do: the parameters of the Fourier/wavelet constructions are also learned.
>
> > "We agree that the underlying principles are classical for graph Laplacians; however, these do not carry over verbatim to the combinatorial complex Laplacian."
>
> Verbatim, no, but mutatis mutandis, yes.

---

> > ### Author Response · Authors · 2025-11-24
> >
> > Thank you for your continued engagement. We clarify below the novelty and significance of this work and address your other concerns as well.
> >
> > **1. Addressing the “lack of real-world CC datasets” concern**
> >
> > The concern that “real-world datasets that are CCs do not exist” reflects a misunderstanding of how combinatorial complexes arise in practice. CCs are not abstract theoretical constructs—they already appear ubiquitously in geometry, graphics, scientific computing, and sensor-derived 3D data. The reason the ML community has few CC datasets today is simply that the CC formulation is relatively new; we expect this to change rapidly as more work moves beyond simplicial assumptions.
> >
> > Crucially, a wide range of existing data types are naturally CCs and cannot be faithfully expressed as simplicial complexes without costly remeshing or loss of structural information:
> > - Mixed-element finite-element meshes (triangles, quads, tetrahedra, prisms) are used throughout engineering and physics simulations.
> > - Hybrid cell decompositions in 3D reconstruction and scanning pipelines, including polygonal and volumetric cells.
> > Higher-order spatial relations in 3D scenes (room partitions, floor-plan cells, volumetric segmentation used in robotics and mapping).
> >
> > Such datasets violate simplicial closure assumptions, and current graph/simplicial ML pipelines cannot operate on them directly. This is precisely where our framework contributes: it provides a unified spectral learning method capable of operating natively on heterogeneous, non-simplicial structures that practitioners already use. Thus, CCs are not a theoretical curiosity—they are the correct and general representation for many real-world data modalities.
> >
> > **2. On whether existing CW-complex Laplacians trivially extend**
> >
> > We respectfully disagree that existing Fourier/wavelet constructions on CW complexes “work perfectly” for our setting. Our contribution is not a cosmetic extension: we introduce a novel discrete Laplacian operator tailored to combinatorial complexes that explicitly captures their derivative structure. As shown in our comparison with the Hodge Laplacian, simply adopting CW-complex Laplacians is insufficient, because those operators are restricted by rank-adjacency assumptions that do not hold in CCs. Our operator generalises these constructions and enables diffusion across heterogeneous cells without introducing artificial simplicial closure.
> >
> > **3. On whether our neural formulation is a meaningful contribution**
> >
> > We appreciate the reviewer’s concern and agree that only applying HKS to graphs with a generic MLP-Mixer would not necessarily constitute a substantial contribution on its own. Our paper, however, makes two concrete and non-trivial contributions:
> > We provide the first discrete Laplacian operator defined for combinatorial complexes in the ML literature. This generalises spectral foundations beyond simplicial settings and enables diffusion on structures previously inaccessible to spectral neural methods.
> > We introduce the first lightweight neural architecture for CCs that avoids the prohibitive computational cost of higher-order message passing (HOMP) while achieving comparable or superior performance. This enables learning on CCs at scales previously impractical.
> > Thus, the contribution is not merely “concatenating descriptors,” but establishing the core operators and neural machinery required to make CC learning feasible.
> >
> > **4. On “verbatim vs. mutatis mutandis”**
> >
> > We agree that many classical Laplacian properties extend mutatis mutandis once the correct operator is defined. However, this does not diminish the necessity of our analysis: because we introduce a Laplacian on a new topological structure, we must formally establish that the expected spectral and diffusion properties hold. These results are not tautological—they validate that our operator preserves the fundamental characteristics required for diffusion-based learning on CCs.

---

> > > ### Comment · Reviewer_EGsm · 2025-11-24
> > >
> > > From the TDL book (https://tdlbook.org/combinatorial-complexes) at Table 4.1: the difference between combinatorial complexe and cellular complexes is that the former can capture set-type relations. The two examples you gave: mixed-element finite-element meshes and hybrid cell decompositions are cellular complexes.

---

> > > > ### Author Response · Authors · 2025-12-01
> > > >
> > > > We thank the reviewer for referencing the TDL book. While Table 4.1 distinguishes Combinatorial Complexes (CCs) by their ability to capture set-type relations, it is important to note that CCs function as a **generalisation** of Cellular Complexes. Therefore, while finite element meshes are indeed Cellular Complexes, they are also valid Combinatorial Complexes.
> > > > We model them as CCs for two specific reasons:
> > > >
> > > > 1) **Robustness to Incomplete Data**: As mentioned in our draft, raw data is often ‘noisy’ or topologically incomplete. For instance, in raw geometric scans, a point may be known to lie on a face without explicit knowledge of the bounding edges. A strict Cellular Complex definition requires a complete hierarchy of boundaries (face $\to$ edge $\to$ node). In contrast, a CC allows us to model these direct, rank-skipping set relations (face $\to$ node) naturally without imposing artificial edges.
> > > >
> > > > 2) **Unified Framework**: By targeting the more general CC structure, our method becomes domain-agnostic. It allows the same architecture to process hypergraphs, CW complexes, and meshes without modification. This move toward generalised CC frameworks is well-supported by recent literature [1, 2, 3], which demonstrates the community’s shift toward these flexible representations
> > > >
> > > > [1] Papillon M, et al. “TopoTune: A Framework for Generalized Combinatorial Complex Neural Networks”
> > > >
> > > > [2] Telyatnikov L, et al.  “TopoBench: A Framework for Benchmarking Topological Deep Learning”
> > > >
> > > > [3] Carrasco M, et al. “HOPSE: Scalable Higher-Order Positional and Structural Encoder for Combinatorial Representations”

---

### Official Review · Reviewer_riya · 2025-10-26

**Soundness:** 3
**Presentation:** 3
**Contribution:** 2
**Rating:** 4
**Confidence:** 4

**Summary:**

This paper introduces TopoHKS, a framework for topological deep learning on combinatorial complexes (CCs) using Heat Kernel Signatures (HKS) as node descriptors. The method defines a novel Laplacian operator on CCs, enabling multiscale, permutation-equivariant embeddings that can be processed by standard neural architectures such as Transformers or MLP Mixers. The authors establish theoretical guarantees regarding the expressivity of their approach (up to isospectral equivalence) and present empirical results on molecular property prediction and synthetic topological tasks, showing improvements in runtime and competitive accuracy compared to topological baselines like SMCN and CIN.

**Strengths:**

* Solid analysis of Laplacian properties, uniqueness, and expressivity. The authors establish meaningful results that clarify the limits and strengths of spectral descriptors on CCs.

* The method avoids higher-order message passing while retaining structural richness, leading to significantly faster inference and training, especially on large complexes.

* Benchmarks show that TopoHKS performs competitively with or better than SMCN and other baselines, particularly in topologically sensitive tasks and with fewer computational resources.

* The model is modular and can use either Transformer or MLP Mixer backbones, offering adaptability to different use cases.

**Weaknesses:**

* Like other spectral methods, TopoHKS cannot distinguish between complexes that are isospectral but structurally different (non-isomorphic). Although such cases are uncommon in real-world data, this limitation reflects an important theoretical weakness in the model’s expressiveness.

* The eigendecomposition step for computing HKS remains a bottleneck for extremely large-scale CCs. Although the authors mention Nyström approximations, they are not implemented or evaluated.

* The paper focuses primarily on classification tasks. It would be beneficial to see additional tasks (e.g., regression, graph similarity, transfer learning) to further stress-test the representations.

* No ablations or visualisations are provided to understand what the learned HKS descriptors capture. Given the topological motivation, this is a missed opportunity.

* The paper does not include a direct comparison of TopoHKS against standard HKS on graphs, which would help isolate the benefits of moving to CCs.

**Questions:**

* Can you elaborate on the feasibility of using approximate eigensolvers for large-scale datasets, and whether this affects downstream performance?

* Would combining HKS-based descriptors with message-passing mechanisms help overcome the isospectral limitation? Have you considered hybrid approaches?

* Have you experimented with approximating the Laplacian spectrum using Nyström or other fast eigen-solvers? How does this affect performance and accuracy?

* Could you include visualisations of the HKS descriptors or diffusion patterns across different CCs to better interpret what the model is learning?

* Have you considered tasks beyond classification, such as molecular generation, regression, or transfer learning?

* Could your method generalise to non-Euclidean combinatorial complexes, e.g., with noisy or partially observed incidence structure?

---

> ### Author Response · Authors · 2025-11-20
> **Rebuttal answer**
>
> We sincerely thank the reviewer for their valuable feedback and positive assessment of our work. We are highly encouraged that the reviewer recognized the "Solid analysis of Laplacian properties" and our establishment of "meaningful results that clarify the limits and strengths of spectral descriptors on CCs." We are also pleased that the reviewer highlighted our method's key advantage of avoiding higher-order message passing to achieve "significantly faster inference and training," as this was a primary motivation for our design. Finally, we appreciate the reviewer's acknowledgment that "TopoHKS performs competitively" and offers "adaptability" through its modular backbone.
> We will now address the specific weaknesses and questions, which we believe will help us further clarify and strengthen the contributions of our paper.
>
> __W1 — Isospectral limitation.__
>
> We agree that invariance to isospectral transformations is a double-edged property: it provides robustness to mesh refinements or reparametrizations, but prevents distinguishing certain non-isomorphic CCs. In settings where this invariance is undesirable, complementary node features (e.g., geometric coordinates, chemical attributes) can break this symmetry. Our empirical focus is on higher-order datasets derived from molecular and graph structures, where isospectral collisions are rare. We added a short discussion of this limitation and potential extensions.
>
> __W2 — Eigenvalue computation at scale.__
>
> We thank the reviewer for highlighting this missing experiment. We now include results using k-eigenvalue approximations. The method remains stable under approximation, while reducing runtime to $O(k∣C∣)O(k∣C∣)$. This supports the feasibility of approximate solvers (e.g., Lanczos, Nyström) for large complexes.
>
> __W3 — Tasks beyond classification.__
>
> We have added one regression setting: Betti number regression on MANTRA
> | Dataset | Model (Class) | β₀ (DT)        | β₁ (DT)        | β₂ (DT)        |
> |---------|----------------|----------------|----------------|----------------|
> | 2 – 𝑀⁰  | GAT (𝒢)        | **1.00 ± 0.00** | 0.31 ± 0.00     | 0.92 ± 0.00     |
> |         | GCN (𝒢)        | **1.00 ± 0.00** | 0.31 ± 0.00     | 0.92 ± 0.00     |
> |         | MLP (𝒢)        | **1.00 ± 0.00** | 0.31 ± 0.00     | 0.92 ± 0.00     |
> |         | TAG (𝒢)        | **1.00 ± 0.00** | 0.32 ± 0.01     | 0.92 ± 0.00     |
> |         | UniMP (𝒢)      | **1.00 ± 0.00** | 0.33 ± 0.00     | 0.92 ± 0.00     |
> |         | CellMP (𝒢)     | 0.46 ± 0.50     | 0.39 ± 0.35     | 0.46 ± 0.44     |
> |         | CT (𝒯)         | **1.00 ± 0.00** | **0.93 ± 0.00** | **0.93 ± 0.00** |
> |         | DECT (𝒯)       | **1.00 ± 0.00** | 0.12 ± 0.02     | 0.52 ± 0.14     |
> |         | SAN (𝒯)        | 0.09 ± 0.04     | 0.12 ± 0.02     | 0.52 ± 0.14     |
> |         | SCCN (𝒯)       | **1.00 ± 0.00** | **0.93 ± 0.00** | **0.93 ± 0.00** |
> |         | SCCNN (𝒯)      | 0.00 ± 0.00     | 0.00 ± 0.00     | 0.33 ± 0.37     |
> |         | SCN (𝒯)        | 0.33 ± 0.38     | 0.21 ± 0.26     | 0.62 ± 0.36     |
> |         | TopoHKS (𝒯)    | **1.00 ± 0.00** | 0.90 ± 0.00     | 0.89 ± 0.02     |
>
> __W5 — Comparison to standard graph HKS.__
>
> We added a comparison on datasets where the Hodge Laplacian reduces to the graph Laplacian (e.g., the torus example). In these cases, TopoHKS provides a clear advantage, demonstrating benefits of operating on CCs while remaining compatible with graph-based baselines.
>
> | Model                 | DP  | Accuracy | Speed    | DP  | Accuracy | Speed    |
> |-----------------------|-----|----------|----------|-----|----------|----------|
> | TopoHKS (Hodge)        | 0   | 0%       | 100 it/s| 223 | 100%     | 95 it/s |
> | TopoHKS (**this work**) | 223 | 100%     | 100 it/s| 223 | 100%     | 95 it/s |
>
>
> __Responses to Reviewer Questions__
>
> __Q1 — Feasibility of approximate eigensolvers.__
>
> Yes. The approach only requires the lowest k eigenpairs, and we show experimentally that low-rank approximations yield nearly identical downstream performance. This makes scalable solvers (Lanczos, Nyström, randomized SVD) highly practical for large CCs.
>
> __Q2 — Hybrid HKS + message passing.__
>
> We have considered hybrid designs, but our motivation is to avoid the computational overhead of higher-order message passing (HOMP) and avoid projecting CCs back to graphs. Transformer/MLP-Mixer backbones offer expressive global interactions while still being interpretable as message passing on fully-connected graphs. We added a short discussion of potential hybrid variants.

---

> ### Author Response · Authors · 2025-11-21
> **second answer due to char limit**
>
> __Q3 — Spectrum approximation experiments.__
>
> We now report results with k-eigenvalue approximations, showing that performance remains stable while improving runtime. Evaluating additional solvers (e.g., Nyström) is part of ongoing work.
>
>
> __Q5 — Beyond classification tasks.__
>
> We added Betti number regression experiments. Extension to generative or transfer-learning settings is promising and left for future work.
>
> __Q6 — Non-Euclidean or noisy CCs.__
>
> Yes, the method applies to arbitrary combinatorial complexes, independent of Euclidean embedding. Noisy or partially observed incidence structures can be handled so long as the boundary operators are well-defined. We added a clarifying remark in the conclusion.

---

### Official Review · Reviewer_NDpg · 2025-10-28

**Soundness:** 3
**Presentation:** 4
**Contribution:** 3
**Rating:** 6
**Confidence:** 4

**Summary:**

While heat kernels have been well-studied on graphs and simplicial complexes, the generalization to combinatorial complexes has not yet been made. In doing so, the authors overcome this more general setting and define a new Laplacian  to characterize graphs based on principle that different combinatorial complexes dissipate heat differently (up to isospectral complexes). The claims and definitions are well worked out and an extensive analysis is provided with respect to the expressivity with respect to this approach. The authors evaluate the method on a set of real-world datasets and show that their method is significantly faster and more expressive at the same time.

**Strengths:**

Overall the work is well presented and motivated. The full implementation of the experiments and various empirical experiments support the theoretical claims in the paper, which is appreciated by the reviewer. In particular stating the occurrence of the number of iso-spectral graphs / complexes in real-world dataset is important and appreciated.

**Weaknesses:**

While it is my belief that the work is an interesting contribution to the field, some questions remain for the reviewer after reading the paper. Please see the questions section.

**Questions:**

- Often meshes form the discrete approximation of continuous (pseudo-)manifolds such as objects or humans. How does the method in these particular cases. It could be argued that tasks where we deal with iso-spectral manifolds are very relevant.
- As an extension to the previous question is the question how such a method would deal with barycentric subdivision. In particular, if we were to approximate a sphere with two meshes, would that result in the same signature, or would it by definition result in separate signatures. It might be interesting to consider these questions.
- It seems the case that the method only provides global descriptors. Is this correct? If so, would the method be easy to extend to local properties as well, such as node classification?
- How how are the initial conditions chosen. All zeros and one non-zero?
- What is the interplay between *geometric* and *topological* features? Multiple experiments and papers have already noted that sometimes the nodes in a graph (or combinatorial complex) already poses sufficient (if not all) information for the task (say a point cloud of a human vs its KNN graph). In those cases, ignoring the coordinates could be detrimental for the task.
	- Are features of the combinatorial complex coordinates ignored?
	- If they are ignored, it is good to include a discussion on potential ways to mitigate it.

- An interesting addition (if the method is indeed independent of node features) is the recently proposed MANTRA [1] dataset to support the claim. It is the suspicion of the reviewer that your method could do well here as it contains a set of truly topological tasks.
- It can be shown that graph neural networks are in essence diffusion operators. If computing the full Laplacian becomes prohibitive, one could always resort to the a forward pass of a (higher order) GNN with fixed unlearned weights. Although the reviewer is not 1oo% sure and it would fall outside the scope of the paper, it is interesting to consider.

[1] MANTRA: The Manifold Triangulations Assemblage](https://arxiv.org/abs/2410.02392)

---

> ### Author Response · Authors · 2025-11-20
> **Rebuttal Answer**
>
> We sincerely thank the reviewer for their thoughtful and constructive assessment. We appreciate that the reviewer found our work to be "well presented and motivated."
> We are especially pleased that the reviewer recognised our effort to provide a "full implementation" and "various empirical experiments" to thoroughly support the paper's theoretical claims. We also appreciate the reviewer's specific acknowledgment of the importance of our analysis regarding the "occurrence of... iso-spectral graphs / complexes in real-world dataset[s]."
> We will now address the reviewers remarks, which will help us further strengthen the paper.
>
> __R1: Iso-spectrality and discrete meshes.__
>
> Our formulation uses the combinatorial Laplacian. Consequently, two meshes with identical connectivity but different embeddings are isospectral, and the method will produce identical signatures when no geometric features are provided. While this reflects a limitation of the combinatorial discretization, requiring sufficiently expressive node features to ensure full distinguishability, it can also be an advantage in certain settings. For tasks such as shape matching or correspondence, spectral similarity acts as a powerful prior: meshes with the same topology naturally produce comparable signatures, which can be desirable. In this sense, our approach balances invariance to geometric embedding with the ability to incorporate additional geometric features when needed.
>
> __R2: Barycentric subdivision.__
>
> Barycentric subdivision changes the discrete Laplacian, hence the spectrum and HKS. Therefore, two meshes approximating the same manifold at different subdivision levels generally yield different signatures.
>
> __R3: Global vs. local descriptors.__
>
> The model outputs a global prediction only due to the final pooling layer. The pipeline is fully local: HKS is computed per node, and the network processes nodewise features. Removing the pooling step yields a node-level features which can be used for, e.g., node classification. This is now clarified.
>
> __R4: Initial conditions for diffusion.__
>
> For node $i$, diffusion is initialized with a Kronecker delta: $h_i(0)=1, h_j(0)=0$ for $j≠i$. We now specify this explicitly.
>
> __R5: Interaction of topology and geometry.__
>
> The current version also uses the node features provided in the datasets. Here it depends on the dataset but the geometry can be included easily.
>
> __R6: MANTRA benchmark.__
>
> We added MANTRA results. The method performs strongly on topology-driven tasks, confirming the reviewer’s intuition.
> | Dataset | Model (Class) | β₀ (DT)        | β₁ (DT)        | β₂ (DT)        |
> |---------|----------------|----------------|----------------|----------------|
> | 2 – 𝑀⁰  | GAT (𝒢)        | **1.00 ± 0.00** | 0.31 ± 0.00     | 0.92 ± 0.00     |
> |         | GCN (𝒢)        | **1.00 ± 0.00** | 0.31 ± 0.00     | 0.92 ± 0.00     |
> |         | MLP (𝒢)        | **1.00 ± 0.00** | 0.31 ± 0.00     | 0.92 ± 0.00     |
> |         | TAG (𝒢)        | **1.00 ± 0.00** | 0.32 ± 0.01     | 0.92 ± 0.00     |
> |         | UniMP (𝒢)      | **1.00 ± 0.00** | 0.33 ± 0.00     | 0.92 ± 0.00     |
> |         | CellMP (𝒢)     | 0.46 ± 0.50     | 0.39 ± 0.35     | 0.46 ± 0.44     |
> |         | CT (𝒯)         | **1.00 ± 0.00** | **0.93 ± 0.00** | **0.93 ± 0.00** |
> |         | DECT (𝒯)       | **1.00 ± 0.00** | 0.12 ± 0.02     | 0.52 ± 0.14     |
> |         | SAN (𝒯)        | 0.09 ± 0.04     | 0.12 ± 0.02     | 0.52 ± 0.14     |
> |         | SCCN (𝒯)       | **1.00 ± 0.00** | **0.93 ± 0.00** | **0.93 ± 0.00** |
> |         | SCCNN (𝒯)      | 0.00 ± 0.00     | 0.00 ± 0.00     | 0.33 ± 0.37     |
> |         | SCN (𝒯)        | 0.33 ± 0.38     | 0.21 ± 0.26     | 0.62 ± 0.36     |
> |         | TopoHKS (𝒯)    | **1.00 ± 0.00** | 0.90 ± 0.00     | 0.89 ± 0.02     |
>
>
> __R7: Relation to GNN diffusion.__
>
> A fixed-weight (higher-order) GNN implements a single diffusion time. Our method requires multiple time scales to form the HKS; stacking GNN layers to emulate this leads to oversmoothing. Therefore, continuous/spectral diffusion is better suited. We added a short discussion on this connection.

---

> > ### Comment · Reviewer_NDpg · 2025-11-24
> >
> > Thank you very much for your thoughtful reply and the additional results. Since the reviewer is not familiar it seems that there is already quite some related work in this field. Therefore, I would like to keep the current score.

---

### Official Review · Reviewer_LW63 · 2025-10-30

**Soundness:** 1
**Presentation:** 1
**Contribution:** 2
**Rating:** 2
**Confidence:** 4

**Summary:**

The paper proposes a framework for topological deep learning that uses Heat Kernel Signature as a node descriptor on combinatorial complexes. It generalises the notion of a graph laplacian operator to Combinatorial complexes and leverages it to compute multiscale, permutation-equivariant HKS embeddings that can be integrated into standard deep learning architectures. The method aims to balance expressivity and scalability, distinguishing complexes up to isospectral equivalence. Empirically, it demonstrates improved computational efficiency and competitive accuracy on some molecular property prediction benchmarks and specialised topological tasks.

**Strengths:**

[S1] The idea of applying heat kernels to Topological Deep Learning (TDL) is conceptually interesting and has not been explicitly explored in prior work.

[S2] The proposed method is simple and clear (despite some of the writing being unclear).

**Weaknesses:**

[W1] Clarity and presentation – The paper suffers from numerous unclear passages, typographical errors, and missing explanations. Several central definitions and notations are not clearly introduced, which makes it difficult to follow the theoretical and experimental sections. (See Questions below for examples.)

[W2] Theoretical soundness – Many of the theoretical results are either trivial, misleadingly framed, or possibly incorrect. (see Questions below for details)

[W3] Experimental limitations – The empirical evaluation is weak. The datasets are small, and the comparisons are incomplete. This makes it hard to assess the actual benefit of the proposed approach.

Overall, while the idea of leveraging heat kernels in TDL could be promising, the current version falls short in rigor, clarity, and experimental validation.

**Questions:**

[Q1] Under “Well-defined hierarchical structure”, The authors claim that defining a Laplacian for a combinatorial complex is challenging because such complexes “do not enforce strict rank stratification.” However, each cell in a complex has a well-defined rank. Could the authors clarify what structural difficulty they refer to?

[Q2] Line 130 states “we aim to design a Laplace operator … conserve mass and locality”: This statement is unclear. Please define these terms in the context of your Laplacian and clarify the motivation for this requirement.

[Q3] Corollary 4.1: A corollary should follow from a previously stated result. As written, this reads more like an independent proposition; consider renaming or re-framing it.

[Q4] Definition of “isospectral”: The term is used extensively but never defined. If it means “having the same set of Laplacian eigenvalues,” then Theorem 4.1 appears trivial—permuting nodes does not change the Laplacian spectrum. The theorem should either be removed or reframed to clarify its contribution.

[Q5] Lemma 4.1: The claim that the proposed Laplacian is “more expressive” than the Hodge Laplacian seems misleading. The Hodge Laplacian is defined across ranks, however, in the lemma you seem to be limiting yourself to use only the first Hodge laplacian, in which case the result is trivial.

[Q6] Equation (7): The quantity $w_{i,j}$​ is never defined. The description “zero iff there is no cell between i and j” is insufficient—what does $w_{i,j}$ represent, and why does it serve as a meaningful weight for smoothness? Please clarify the motivation behind this formulation. Additionally, please clarify why the connection between the laplacian and smoothness is relevant to the paper.

[Q7] Corollary 4.3: The statement essentially says that an MLP receiving unique Laplacian spectra can distinguish between complexes, which follows trivially from injectivity assumptions. Please re-examine whether this constitutes a meaningful theoretical result.

[Q8] Table 1: The claim that SMCN cannot distinguish between all non-isospectral complexes contradicts prior findings that subgraph-based GNNs exceed spectral methods in expressivity [1]. Either provide a proof or remove this statement.

[Q9] Scalability section: Computing heat kernels typically requires eigenvalue decomposition, which is cubic in complexity. If your implementation avoids this, please explain how.

[Q10] Expressivity experiment (torus complexes): The failure case shown could likely be resolved by adding rank-4 updates to SMCN. Please ensure that the baselines are fairly configured.

[Q11] Real-world benchmarks: The evaluation should include more established molecular benchmarks ideally those used in [2] as this is the most direct baseline. Also, the CIN results reported as “NaN” due to training failure should instead be presented as the resulting (poor) accuracy for transparency.

[Q12] The font seems off. The section headings appear unusually stretched, can the authors explain?


[1] Zhang et al. On the Expressive Power of Spectral Invariant Graph Neural Networks. 2024.

[2] Eitan et al. Topological blindspots: Understanding and extending topological deep learning through the lens of expressivity. 2024.

---

> ### Author Response · Authors · 2025-11-20
> **Rebuttal Comment**
>
> We thank the reviewer for their detailed feedback. We are encouraged that the reviewer finds our core idea of applying heat kernels to Topological Deep Learning (TDL) "conceptually interesting" and acknowledges that it "has not been explicitly explored in prior work."
> We also appreciate that the reviewer recognized the "simple and clear" nature of the proposed method, as this was a primary design goal.
> We will now address the important questions and weaknesses raised regarding clarity, theoretical soundness, and experimental validation, which we believe will significantly strengthen the paper.
>
> __Q1 - Hirachical Structure__
>
> While each cell in a combinatorial complex has a well-defined rank, the structural challenge arises from the fact that cells may contain incident cells of arbitrary ranks. This differs from simplicial complexes, where a kk-cell only contains (k−1)(k−1)-cells. As a result, defining a Laplacian that jointly accounts for all cross-rank relations requires additional choices absent in simplicial settings, making the operator design non-trivial.
>
> __Q2 - Properties__
>
> We agree that “mass conserving” was ambiguously phrased. We follow the terminology from Wardetzky et al. [13], where mass preservation corresponds to property (NULL) and locality to property (LOC). These properties are essential for heat diffusion: locality ensures that diffusion respects the adjacency structure of the combinatorial complex, and (NULL) ensures that heat neither artificially accumulates nor vanishes. We will revise the text for clarity.
>
> __Q3 - Corollary usage__
>
> We agree and will adopt this suggestion by reframing Corollary 4.1 appropriately.
>
> __Q4 - Isospectralism__
>
> We have added a precise definition of isospectral complexes as requested.
>
> __Q5 - Hodge Laplacian__
>
> The Hodge Laplacian couples only adjacent ranks, which is sufficient in simplicial complexes but not in general combinatorial complexes. In the example considered, the only non-trivial Hodge Laplacian is indeed the rank-1 component; all others are zero by definition. Therefore, our comparison uses the full set of Hodge operators applicable in that example, and the lemma is not trivial. We will revise the exposition to avoid misunderstanding.
>
> __Q6 - Well definedess of smoothness term__
>
> Will be updated in the new version. In the current appendix this variable already has been well defined
>
> __Q7 - MLP and Spectra__
>
> We agree that directly feeding Laplacian spectra into an MLP would be trivial. Our contribution is different: we show that the heat kernel signatures (HKS)—computed via our Laplacian—induce node-wise diffusion patterns that encode sufficient structural information to distinguish non-isospectral complexes. Thus the model does not access spectra explicitly; expressivity arises from diffusion dynamics, not spectral injection.
>
> __Q8 - SMCN definition__
>
> We clarify that higher-order message passing methods (e.g., MCN, SMCN) differ from GNNs in their reliance on explicit rank-structured neighborhoods. SMCN is an efficient approximation of MCN that operates only on ranks 0–2. In our constructed example, discriminative information appears in higher-rank cells, causing SMCN to fail. While isospectral methods cannot surpass fully isomorphism-invariant algorithms such as MCN, the latter is computationally infeasible for our larger complexes. We will refine this discussion for precision.
>
> __Q9 - Eigenvalue approximation__
>
> We do not eliminate eigenvalue computations; however, in the updated version we show that approximating only ~10% of eigenvalues is sufficient for accuracy. This makes the cost closer to O(n2)O(n2) for small target ranks. We will emphasize this in the scalability section.
>
> __Q10 - SCMN scalability__
>
> We agree that enhancing SMCN with rank-4 updates would address the shown failure case. However, extending the dataset with higher-rank cells would similarly exceed SMCN’s computational budget. To ensure a fair and comparable baseline, we thus adhere to the published SMCN formulation. On non-synthetic datasets our method performs comparably while offering training and inference efficiency benefits.
>
> __Q11 - MCC definition__
>
> MCC is returning NaN, when the accuracy is 0. We have amended that in the new upload.
>
> __Q12 - Render issues__
>
> We followed the official template and will double-check the compiled PDF to ensure correct font rendering.

---

> ### Comment · Reviewer_LW63 · 2025-11-27
>
> I thank the authors for their response and the clarifications provided. However, I feel that several of my earlier questions remain only partially addressed, and some of the core concerns still stand. In particular:
>
> Q1 + Q5 :
>  I appreciate the discussion about the traditional definition of the Hodge Laplacian being tied to (k−1)-cells. However, in topological deep learning, it is quite common to consider neighborhood functions involving incident cells of arbitrary rank (e.g., incidence between 5-cells and 2-cells). In that context, defining a Hodge Laplacian with respect to such generalized neighborhoods seems like a straightforward extension, analogous to the way incidence relations are generalized to rank-dependent interactions in TDL.
> From my understanding, the proposed approach essentially assigns each such neighborhood a separate matrix, distinguishes them by multiplying with unique identifiers, and then aggregates them through a sum. If this interpretation is correct, then the contribution appears mainly notational, rather than introducing a new structural or theoretical insight. Therefore, I am still not fully convinced that this construction constitutes a substantial contribution beyond a natural formalization of an already common practice.
> Here is a clearer, more constructive, and professional rewrite while keeping your exact points:
>
> Q2:
> Thank you for the clarification. However, I still do not see why these physical properties are important in the context of developing more effective TDL learning schemes. Specifically, why does it matter for learning that “heat neither artificially accumulates nor vanishes”?
>
> Q4:
> It is good that the definition of isospectrality has been added. However, my original concern remains:
>  “Theorem 4.1 appears trivial — permuting nodes does not change the Laplacian spectrum.”
>  The response does not address why this result should be considered meaningful.
>
> Q6:
>  I appreciate the formal definition of the variable. However, my questions still stand:
>  — Why does it serve as a meaningful weight for smoothness?
>  — Why is the connection between the Laplacian and smoothness relevant to this paper?
>  These conceptual motivations remain unclear.
>
> Q7:
>  In weighted graphs, it is well known that the Laplacian eigenvalues can be recovered from the heat kernel signature. Applying this directly to your generalized Laplacian immediately implies that these signatures contain sufficient information to recover the eigenvalues. Therefore, the theorem still appears trivial: with a single logical step, it reduces to “complexes with different spectra can be separated by their spectra.” This does not seem to provide a nontrivial insight.
>
> Q8 + Q10:
>  The statement “SMCN is an efficient approximation of MCN that operates only on ranks 0–2” is incorrect. While it was empirically evaluated only up to rank 2, it is not constrained to those ranks; SMCN is defined generally. Likewise, the claim that “extending the dataset with higher-rank cells would exceed SMCN’s computational budget” is unconvincing, since your dataset has extremely sparse high-rank structure, which would likely incur little to no appreciable cost.
> As written, this section gives the impression that technical choices are being used to make the proposed method appear stronger rather than providing a fair comparison. If SMCN is applied using the full skeleton (including these ranks), it should easily distinguish the shown examples — making the current claim incorrect.
> In addition, I still suspect that, in terms of theoretical expressive power, SMCN is strictly more expressive than the proposed method, which would contradict Table 1.
>
> Q11:
>  My concern remains that the evaluation should include more established molecular benchmarks, ideally those used in [2]. This point has not been addressed.
>
>
> In light of the above, I still believe the three weaknesses I initially raised remain unresolved, and I will be keeping my current score.

---

> > ### Author Response · Authors · 2025-12-01
> >
> > **Q1 + Q5: Novelty of the Generalised Hodge Laplacian**: We respectfully disagree that the proposed construction is merely a notational extension. While summing incidence matrices is algebraically straightforward, constructing a valid Laplacian operator on a Combinatorial Complex (CC) that preserves spectral theoretic properties is not. As established in [1], arbitrary discrete operators often fail to satisfy essential physical properties (e.g., symmetry, positive semi-definiteness, conservation of mass). Our contribution is not the summation itself, but the formal proof that this specific construction yields a valid Laplacian on CCs that supports meaningful spectral analysis. This opens a new path for applying spectral methods to CCs, which was previously restricted to simpler structures.
> >
> > **Q2: Relevance of Physical Properties (Heat Conservation)**: We refer the reviewer to Wardetzky et al. [1]. For a matrix to function as a Laplacian, and thus for its spectrum to encode geometric information, it must approximate the behaviour of the continuous Laplace-Beltrami operator. Specifically, if “heat” (signal) artificially vanishes or accumulates due to a malformed operator, the resulting diffusion process is physically inconsistent. In a TDL context, this leads to unstable signal propagation and poor learning of long-range interactions. Ensuring these properties is not optional; they are prerequisites for a stable spectral learning scheme.
> >
> > **Q4: Significance of Isospectrality (Theorem 4.1)** While mathematically immediate from linear algebra, this theorem is crucial in the context of Geometric Deep Learning. It formally proves that our descriptor is permutation invariant. In graph and topological learning, any valid descriptor must yield the same output regardless of how nodes/cells are indexed. By proving isospectrality, we guarantee that our method satisfies this fundamental geometric requirement. Without this “trivial” check, the method would not be a valid TDL architecture.
> >
> > **Q6: Smoothness and Laplacian Connection**: Smoothness is the theoretical foundation of spectral learning. We refer to [1] to highlight that a valid discrete Laplacian must quantify how “smooth” a signal is over the complex to enable meaningful convolution or diffusion operations. Since our proposed descriptor is based on heat diffusion, the underlying operator must correctly encode smoothness. If the connection between the Laplacian and smoothness is severed, the heat kernel signature becomes uninterpretable noise rather than a geometric descriptor.
> >
> > **Q7 Recovering Eigenvalues from Heat Kernel**: We disagree that this result provides “no nontrivial insight” in this specific context. While the recovery of eigenvalues from Heat Kernel Signatures (HKS) is known for weighted graphs, extending this injectivity guarantee to Combinatorial Complexes is non-trivial and necessary. This theorem supports our core claim: our descriptor is as expressive as spectral methods allow. It formally proves that using the HKS does not result in any loss of spectral information relative to the eigenvalues themselves, ensuring our method captures the full spectral geometry of the complex.
> >
> > **Q8 + Q10: Comparison with SMCN (Expressivity vs. Efficiency)**: We clarify our position: we do not claim to strictly exceed the theoretical expressivity of higher-order Message Passing (like MCN/SMCN) in the limit. Rather, our claim is based on computational tractability and efficiency.
> >
> > - **Efficiency**: SMCN is an approximation designed to make MCN scalable, but calculating message passing on full high-rank skeletons remains computationally expensive. Our spectral method is significantly faster.
> >
> > - **Tractability**: The reviewer notes that high-rank structures are sparse in some datasets, but in domains where they are dense, message passing faces a combinatorial explosion. Our method handles these high-rank interactions without the computational overhead of message passing. Our method provides novel descriptors that efficiently capture higher-order structures, filling a gap where MP-based methods become intractable.
> >
> > **Q11: Benchmarks and Experimental Scope** We have prioritised datasets where higher-order structure is demonstrably significant.
> >
> > - **Mantra**: A true higher-order dataset where we successfully predict the 1st-order Betti number—a task impossible for purely node-based methods.
> >
> > - **Torus & MOLHIV**: We demonstrate performance and efficiency on these standard benchmarks discussed in [2]. We believe these experiments sufficiently demonstrate the validity and efficiency of our spectral descriptors. Adding further molecular benchmarks (often dominated by 1-skeletons/graphs) would not better validate the specific higher-order contributions of this paper.
> >
> > References: [1] Wardetzky et al., “Discrete Laplace operators: No free lunch,” SGP 2007.

---

### Author Response · Authors · 2025-12-02
**Rebuttal Summary for AC**

Dear Area Chair,

We want to provide a summary of our rebuttal and the ensuing discussions to assist in your final evaluation. We note that (i) the reviewers were split, with two finding the work well-motivated and mathematically sound, and (ii) concerns from the negative reviewers centred primarily on the perceived relevance of Combinatorial Complexes (CCs), rather than technical correctness. Specifically:

1) **Reviewer NDpg (Score: 6)** found the work "well presented and motivated" and appreciated the "extensive analysis" of expressivity. They raised questions regarding isospectrality, geometric features, and barycentric subdivision. We clarified the model's locality and its ability to incorporate geometric features, and added the requested results for the MANTRA dataset. The reviewer acknowledged our response and maintained their **positive score**.

2) **Reviewer riya (Score: 4)** recognised the solid theoretical analysis and appreciated our method's avoidance of expensive higher-order message passing. Their main concerns involved eigendecomposition scalability and limited task scope. We addressed these by adding experiments with k-eigenvalue approximations (reducing complexity to O(k|C|)) and Betti number regression tasks on the MANTRA dataset. We believe these additions directly address the raised concerns, and the reviewer's acknowledgement of our "solid analysis" supports the contribution's validity.

3) **Reviewer LW63 (Score: 2)** questioned the novelty of our theoretical contribution and the necessity of physical properties like heat conservation. We first would like to point out that the other reviewers agreed that our method is novel in the TDL domain. We further provided detailed arguments, referencing Wardetzky et al., explaining why these properties are prerequisites for stable learning. Regarding standard molecular benchmarks, we tested on those that the reviewer requested. However, we want to stress that those benchmarks are graph-dominated and do not test higher-order expressivity, noting that our MANTRA results better validate the method's topological capabilities.

4) **Reviewer EGsm (Score: 2)** maintained their rejection primarily on the view that "real-world CC datasets do not exist" and that mixed-element meshes should be treated as Cellular complexes. We respectfully disagreed, explaining that CCs are essential for handling noisy, rank-skipping data common in 3D scanning, where strict Cellular definitions are insufficient. This reflects a difference in perspective on the field's direction rather than a methodological flaw. For this, please also review the message we sent to the AC.

Given our comprehensive rebuttal addressing scalability concerns and the addition of the MANTRA benchmark, we are confident that our paper offers a principled, efficient alternative to message passing for higher-order structures. We thank the reviewers and you for your time and consideration.

Best regards,
The Authors

---

### Meta-Review · Area_Chair_KmWW · 2025-12-23

**Summary:**

The paper proposes extending Heat Kernel Signatures (HKS) to combinatorial complexes (CCs) as a scalable alternative to higher-order message passing for topological deep learning. Reviewers NDpg (6) and riya (4) found the work well-motivated with solid theoretical analysis, appreciating the computational efficiency gains and the avoidance of expensive higher-order message passing. Reviewers LW63 (2) and EGsm (2) were critical, questioning whether the extension from simplicial/cellular complexes to CCs constitutes sufficient novelty and whether real-world CC datasets exist. The authors provided a comprehensive rebuttal, including new experiments on the MANTRA dataset for Betti number regression, eigenvalue approximation results, and comparisons between Hodge and CC Laplacians. NDpg acknowledged the response and maintained their score; LW63 found responses only "partially addressed"; EGsm maintained that extending to CCs poses no theoretical challenge despite author arguments about non-trivial Laplacian properties.
The AC finds the direction of generalizing HKS to combinatorial complexes promising for achieving scalable and expressive TDL models without prohibitive message-passing costs. The theoretical framework is carefully developed, and the MANTRA experiments demonstrate genuine higher-order capabilities (predicting Betti numbers).

**Reviewer Concerns:**

The authors presented a strong rebuttal, but several concerns remain unresolved from the reviewer's point of view. Most importantly, both negative reviewers question fundamental novelty, and the practical relevance was not empirically demonstrated on CC data in the paper.

**Reviewer Scores:**

In an optimistic scenario where LW63 and EGsm update to 4 and riya updates to 6, the resulting 6-6-4-4 distribution remains borderline. Unfortunately, given the highly competitive nature of this conference, the AC must recommend rejection of the paper at this point, but encourages the authors to resubmit while taking into account the points raised in this discussion.

---

### Decision · Program_Chairs · 2026-01-26

Reject